# CroPe: Cross-Modal Semantic Compensation Adaptation for All Adverse Scene Understanding

**Qin Xu**[1,2]**, Qihang Wu**[1,2]**, Hongtao Lu**[1,2]**, Xiaoxia Cheng**[1,3]**,**[∗]**Bo Jiang**[1,2∗]

[1]School of Computer Science & Technology, Anhui University
[2]Anhui Provincial Key Laboratory of Multimodal Cognitive Computation, Anhui University
[3]College of Computer Science & Technology, Zhejiang University
`xuqin@ahu.edu.cn, {e23301339, e24301204}@stu.ahu.edu.cn`
`zjucxx@zju.edu.cn, zeyiabc@163.com`
*Project website:* `https://github.com/wqh011128/CroPe`

## Abstract

Scene understanding in adverse conditions, such as fog, snow, and night, is challenging due to the visual appearance degeneration. In this context, we propose a Cross-modal Semantic Compensation Adaptation method (**CroPe**) for scene understanding. Distinct from the existing methods, which only use the visual information to learn the domain-invariant features, CroPe establishes a visual-textual paradigm which provides textual semantic compensation for visual features, enabling the model to learn more consistent representations. We propose the Complementary Perceptual Text Generation (CPTG) module which generates a set of multi-level complementary-perceptive text embeddings incorporating both generalization and domain awareness. To achieve cross-modal semantic compensation, the Reverse Chain Text-Visual Fusion (RCTVF) module is developed. By the unified attention and reverse decoding chain, compensation information is successively fused to the visual features from the deep (semantic dense) to shallow (semantic sparse) features, maximizing compensation gain. CroPe yields competitive results under all adverse conditions and significantly improves the state-of-the-art performance by 6.5 mIoU for ACDC-Night dataset and 1.2 mIoU for ACDC-All dataset, respectively.

## 1 Introduction

Scene understanding under adverse weather conditions serves an essential task for outdoor applications, such as autonomous driving, surveillance systems, and disaster response. However, due to the extreme changes in illumination, texture, and occlusion patterns under various adverse conditions, the large domain discrepancies across diverse scenes pose a significant challenge, making it difficult for existing methods to effectively address segmentation under all adverse weather conditions.

Existing methods can be divided into two groups. One is scene-specific framework [1, 2, 3, 4, 5, 6], which is tailored to particular adverse conditions. For example, BWG [3] enhances the generalization ability for foggy scenes through content enhancement and style decorrelation. S2R2 [2] jointly optimizes deraining and segmentation tasks using contrastive learning. Despite the successes in certain scenarios, the model's performance declines when confronted with more complex and diverse scenes. Another group is scene-agnostic framework [7, 8, 9, 10, 11, 12] which offers a more unified solution. For instance, PASS [13] utilizes an implicit visual prompt strategy to enhance cross-domain consistency by eliminating domain-specific weather features. MIC [9] captures contextual information of the scene through mask reconstruction. However, whether scene-specific or scene-agnostic

---

[∗]Corresponding authors: Xiaoxia Cheng and Bo Jiang

39th Conference on Neural Information Processing Systems (NeurIPS 2025).

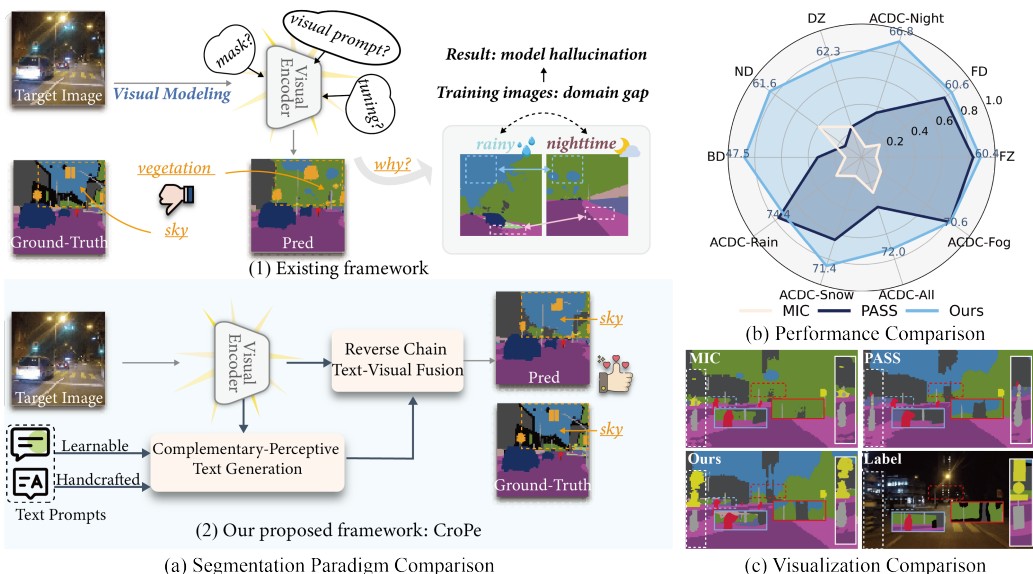

Figure 1: Comparison between our CroPe and existing methods: (a) In contrast to existing methods relying solely on visual modality, our CroPe integrates visual and text modality. (b) Our CroPe outperforms existing methods in many adverse scenarios. (c) Existing methods often produce hallucinations (e.g., incorrectly classifying the sky as trees). We highlight the differences between our CroPe and existing methods using dashed boxes, and zoom in with solid boxes for emphasis.

frameworks, they only train the model within the visual modality (e.g. using visual prompts, mask reconstruction or tuning) to learn domain-invariant knowledge for unsupervised domain adaptation (UDA) segmentation, as shown in Figure 1 (a)(1). Under adverse conditions, visual features often experience severe degradation (e.g., loss of texture, low visibility, and color distortion). This degradation makes the model prone to hallucinations and complicates the learning of consistent semantic knowledge across different domains.

To address the aforementioned issue, we introduce the integration of robust text modality into unsupervised domain adaptive semantic segmentation (UDASS) under adverse conditions and propose a novel cross-modal semantic compensation adaptation method (**CroPe**) for all adverse scene understanding. Since text semantics are invariant to environmental changes and facilitate acquisition of across-domain class-consistent semantics, our CroPe leverages text semantics as a high-level guidance modality to compensate for degraded visual information under adverse conditions, which can effectively obtain semantic consistency across different domains, as illustrated in Figure 1 (a)(2). Specifically, we propose the Complementary-Perceptive Text Generation (CPTG), which is composed of a decoupling strategy, domain-specific perception, domain-invariant regularization, and gated complementary fusion. The decoupling strategy is proposed to decouple the text embeddings into domain invariant embedding which is constrained by domain-invariant regularization and domain perceptive embedding which is interacted with visual features by the domain-specific perception. The gated complementary fusion is designed to adaptively fuse the domain-specific and domain-invariant text embeddings. After the CPTG module, the Reverse Chain Text-Visual Fusion (RCTVF) module is designed, which develops the unified attention and reverse decoding chain. The unified attention mechanism integrates the multi-scale visual features and the multi-level textual features outputted by CPTG. The reverse decoding chain incorporates the visual features compensated with deep semantics into the shallower, yet unfused, visual feature. By this chained fusion, compensation gain of visual features can be maximized. CroPe surpasses most of existing visual frameworks, effectively addressing visual degradation in UDASS, avoids erroneous classifications such as mistaking skies for roads. The performance superiorities of CroPe in comparison with the SOTA methods on ten datasets under challenging scenarios is shown in Figure 1 (b). The segmentation maps presented in Figure 1 (c) qualitatively illustrate CroPe's adaptability in adverse scenarios.

Our contributions are briefly summarized as follows:

- We propose a cross-modal semantic compensation method, which integrates the textual modality into the unsupervised domain adaptation semantic segmentation task under adverse scenes to enhance the model's adaptability.
- We design CPTG module to generate multi-level complementary-perceptive text embeddings, which are then integrated into visual features using the RCTVF module to achieve cross-modal semantic compensation.
- Extensive experimental results show that our method achieves state-of-the-art performance in various adverse scenarios, including rain, snow, fog, and nighttime, while also reducing training cost. This highlights the model's superiority in both effectiveness and efficiency.

## 2  Related Work

Adverse visual scenes hinder effective knowledge transfer in unsupervised domain adaptation (UDA) for scene understanding. Early studies primarily focus on single scenarios. For example, FIFO [14] focuses on fog scenes and learns fog-invariant representations by extracting fog-related factors from style features. Some works focus on night scenes [15, 16], using pseudo-supervision through day-night paired images or cross-temporal correspondences. These methods perform well in specific scenes but struggle to generalize across diverse adverse conditions. Recent research turns to developing a unified framework capable of handling multiple adverse scenes simultaneously [17, 13]. For example, Refign [18] introduces an uncertainty-aware dense matching method to align the target image and the reference image from various adverse scenes, thereby improving its robustness across multiple adverse scenes. DAFormer [7] further improves the expressiveness in various scenarios by introducing training strategies such as Transformer encoder, rare category sampling, and ImageNet feature distance constraints. SePiCo [19] proposes a semantically guided pixel comparison method, which constructs a cross-domain discriminative embedding space through center point-aware and distribution-aware comparison losses, and simultaneously optimizes feature alignment and self-training stability. The unified framework has become the mainstream solution due to its powerful cross-scenario capabilities. However, existing unified framework methods rely on visual modeling to capture domain invariance and struggle to address the challenges posed by significant visual distortion. In this paper, we introduce the first cross-modal semantic compensation method to learn domain-invariant features for UDA. More related works on UDA semantic segmentation can be found in Appendix A.1.

## 3  Method

In this section, we first give the task formulation of an unsupervised domain adaptation (UDA) scene understanding in adverse scenes in §3.1 and then describe our method in detail. As shown in Figure 2, our proposed CroPe consists of two components, a Complementary-Perceptive Text Generation §3.2 and a Reverse Chain Text-Visual Fusion §3.3.

### 3.1  Task Formulation

Given a training sample $(I^S, I^T, y^S)$, where $(I^S, I^T) \in \mathbb{R}^{3 \times H \times W}$ represents the input images of the training set from the source domain $S$ and the target domain $T$, and $y^S \in \mathbb{R}^{H \times W}$ is the corresponding image label from the source domain, $H$ and $W$ represent the resolution. The goal of the UDA scene understanding task is to use $I^S$, $y^S$, and $I^T$ for training a model with good segmentation performance in the target domain test set.

### 3.2  Complementary-Perceptive Text Generation

The Complementary-Perceptive Text Generation (CPTG) aims to obtain a set of multi-level complementary-perceptive text embeddings to provide more effective completion and alignment for cross-modal semantic compensation from textual to visual modality. A key challenge of CPTG is the design of textual prompts, as using learnable or hand-crafted text prompts alone poses the risks of overfitting and limited flexibility, respectively. Therefore, we propose to use learnable prompts as the core and combine them with hand-crafted prompts in the Domain-Invariant Regularization to complement each other.

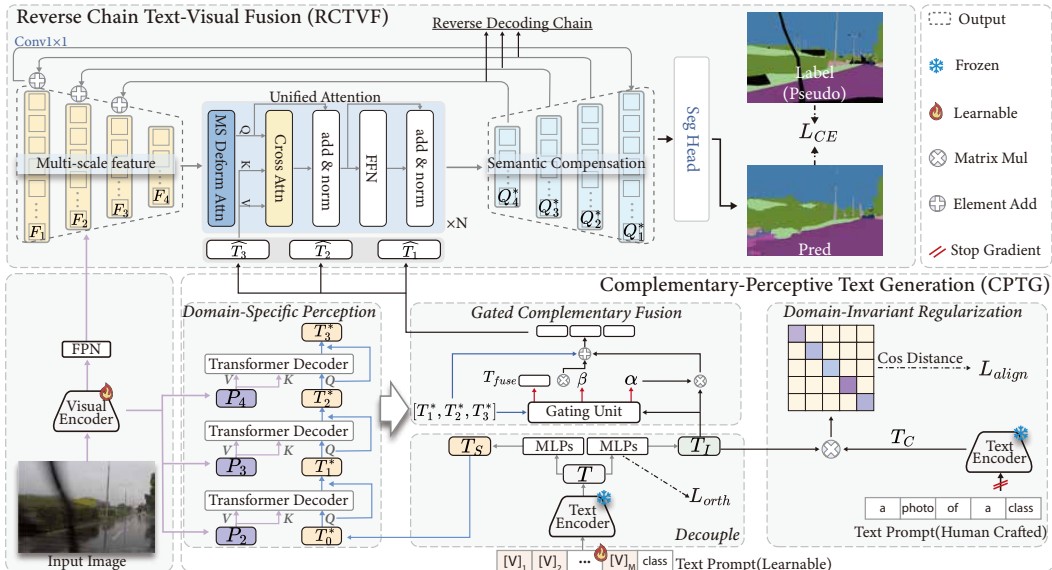

Figure 2: The overview of our proposed CroPe. The CPTG module receives image features $[P_2, P_3, P_4]$ from the visual encoder and global text features $T$ from the text encoder. It processes them through Domain-Specific Perception, Domain-Invariant Regularization, and Gated Complementary Fusion to generate multi-level complementary-perceptive text embeddings $\widehat{T} = [\widehat{T}_1, \widehat{T}_2, \widehat{T}_3]$. Next, the RCTVF module fuses $\widehat{T}$ into the multi-scale visual features $[F_1, F_2, F_3, F_4]$ through Unified Attention and Reverse Decoding Chain, outputs the multi-scale semantic compensated visual features $Q^* = [Q_1^*, Q_2^*, Q_3^*, Q_4^*]$, which are then passed into the segmentation head to obtain the final segmentation predictions.

Specifically, we first input the learnable text prompt of each category $[V]_1[V]_2 \ldots [V]_M[\text{class}]$ into the text encoder to generate the corresponding raw text embeddings $T \in \mathbb{R}^{C \times D}$, where $V$, $[\text{class}]$, $M$, $C$, and $D$ denote the context word, category name, the number of context words, the number of categories, and the dimension of text embedding, respectively.

**Decoupling Strategy.** Instead of direct optimization of $T$, we propose a decoupling strategy to learn two independent structures of textual prompts, i.e., generality and domain awareness. Specifically, we use two identical multi-layer perceptrons (MLPs) (with non-shared weights) to decouple $T$ into the domain-specific embedding $T_S \in \mathbb{R}^{C \times D}$ and the domain-invariant embedding $T_I \in \mathbb{R}^{C \times D}$. Meanwhile, to ensure clear semantic independence between these two decoupled feature embeddings, we apply an orthogonality constraint on $T_S$ and $T_I$. The orthogonal loss is defined as follows:

$$L_{orth} = \|\langle T_S, T_I \rangle\|^2, \tag{1}$$

where $\langle ., . \rangle$ represents the cosine similarity. By explicitly decoupling $T_S$ and $T_I$, we can establish optimization objectives for domain invariance and domain perception independently.

**Domain-Specific Perception.** To enhance the domain-awareness ability of $T_S$, we propose a domain-specific perception module where the correlation of $T_S$ and the visual representation is explored and mined. To balance computational efficiency and the richness of domain-specific information, we use the last three level features $[P_2, P_3, P_4]$ from the visual encoder as the visual clues with the domain perception. Then, both $T_S$ and $[P_2, P_3, P_4]$ are projected from $D$ to the channel dimension $D_d$ through linear layers, with $T_S$ mapped to the query tokens $Q_{T_S}$, and $P_i$ mapped to the key tokens $K_{P_i}$ and value tokens $V_{P_i}$, where $i \in \{2, 3, 4\}$. Through attention interactions, $T_S$ can capture multi-level domain-specific information, the process is expressed as:

$$Q_{T_i^*} = Q_{T_{i-1}^*} + \text{Softmax}\left(\frac{Q_{T_{i-1}^*} K_{P_{i+1}}^\top}{\sqrt{D_d}}\right) V_{P_{i+1}}, i \in \{1, 2, 3\}, \tag{2}$$

where $T_0^* = T_S$. Finally, each output $Q_{T_i^*}$ is restored from $D_d$ to $D$ dimension through a linear layer, obtaining $T^* = [T_1^*, T_2^*, T_3^*] \in \mathbb{R}^{3 \times C \times D}$ as the multi-level perception output. By aligning $T_S$ and

the visual space, we obtain the domain-specific perceptive embedding $T^*$ which captures the local and global context.

**Domain-Invariant Regularization.** To enable the domain-invariant embedding $T_I$ to effectively generalize across different domains, we utilize manually designed prompts that provide general representations and maintain invariance. Concretely, by creating a general text prompt and obtaining the corresponding text embedding $T_C \in \mathbb{R}^{C \times D}$ through the text encoder, we exploit a cosine similarity soft constraint $L_{align}$ between $T_I$ and $T_C$, ensuring that $T_I$ does not deviate excessively from $T_C$, thereby maintaining the domain robustness. The soft constraint $L_{align}$ is defined as follows:

$$L_{align} = 1 - \langle T_I, T_C \rangle. \tag{3}$$

By minimizing the $L_{align}$ in Equation (3), we can both prevent $T_I$ from overfitting to a specific domain and allow $T_I$ to learn more generalized representations.

**Gated Complementary Fusion.** To generate complementary-perceptive text embedding, we propose a gated complementary fusion that integrates domain-specific and domain-invariant text embeddings through dynamically adjusting the contribution of each embedding. Primarily, a Gating Unit is designed. In this unit, $T_I \in \mathbb{R}^{C \times D}$ is broadcasted and concatenated with $T^* \in \mathbb{R}^{3 \times C \times D}$ along the dimension of the feature as the input, and then passed through a linear layer to obtain the fused feature $T_{fuse} \in \mathbb{R}^{3 \times C \times D}$. Meanwhile, we fed the input into a single hidden layer multilayer perceptron (MLP) and a Sigmoid activation function to learn class-specific gating weights $\{\alpha, \beta\} \in \mathbb{R}^{1 \times C \times 1}$ for $T_I$ and $T_{fuse}$. Finally, we obtain the multi-level complementary-perceptive text embeddings $\widehat{T} = [\widehat{T}_1, \widehat{T}_2, \widehat{T}_3] \in \mathbb{R}^{3 \times C \times D}$ as followings:

$$\widehat{T} = T^* + \alpha \cdot T_I + \beta \cdot T_{fuse}. \tag{4}$$

The gated complementary fusion mechanism dynamically adjusts the fusion weights between domain perception and generalization information, enabling the complementary integration of the text semantics. This mechanism effectively enriches text cross-domain representations, providing a robust basis for enhancing the semantic density of visual features.

## 3.3 Reverse Chain Text-Visual Fusion

To fully integrate multi-level complementary-perceptive text embedding semantics into the visual modality and achieve cross-modal semantic compensation, we propose a Reverse Chain Text-Visual Fusion (RCTVF) module. As shown in Figure 2 (Upper), RCTVF receives $\widehat{T}$ generated by the CPTG module and the multi-scale features $[F_1, F_2, F_3, F_4]$ generated by the FPN [20]. The resolution of the multi-scale features is $[\frac{1}{4} \times, \frac{1}{8} \times, \frac{1}{16} \times, \frac{1}{32} \times]$ of $(H, W)$, respectively. RCTVF consists of Unified Attention and Reverse Decoding Chain components.

**Unified Attention.** The Unified Attention integrates multi-scale deformable attention [21] and cross-modal attention, allowing the model to concentrate on key areas at various scales while maintaining text semantic guidance. Then, we fuse the channel information through a feedforward network (FFN) to enhance the semantic density of visual features.

Specifically, we input one of the scales of $F_i (i \in \{1, 2, 3, 4\})$ into the Unified Attention module at a time, initially capturing both detail and global information via the multi-scale deformable attention. Next, we interact the output visual information with the corresponding $\widehat{T}_{i-1}$ using the cross-modal attention, and then further integrate the cross-modal semantics through the FFN. We also map $F_i$ and $\widehat{T}_{i-1}$ to a dimension $D_d$ to create $Q_{F_i}$, $K_{\widehat{T}_{i-1}}$, and $V_{\widehat{T}_{i-1}}$. The specific calculations are as follows:

$$Q_i^* = \text{FFN}(\text{Cross-Attn}(\text{Deform-Attn}(Q_{F_i}), K_{\widehat{T}_{i-1}}, V_{\widehat{T}_{i-1}})), i = 4, 3, 2. \tag{5}$$

Among them, the semantic compensated visual features $[Q_2^*, Q_3^*, Q_4^*]$ of each scale can be generated using Equation (5). Due to $Q_{F_1}$ having the largest resolution, we control the computational complexity by processing $Q_{F_1}$ with $1 \times 1$ convolution to obtain $Q_1^*$.

**Reverse Decoding Chain.** To maximize semantic compensation, we propose the Reverse Decoding Chain. It utilizes a reverse chain from deep to shallow layers for decoding compensation, addressing semantic isolation among the current $Q_i^*$. Specifically, $F_4$ has the densest semantic information in

$[F_1, F_2, F_3, F_4]$, while in the multi-level text embedding $[\widehat{T}_1, \widehat{T}_2, \widehat{T}_3]$, the image region perceived by $\widehat{T}_3$ is more global. Therefore, we first calculate the Unified Attention of $F_4$ and $\widehat{T}_3$ (not parallel processing $[F_1, F_2, F_3, F_4]$), and pass it through the following formula:

$$Q_{F_{i-1}} \leftarrow Q_{F_{i-1}} + \mathrm{Up}(Q_i^*, Q_{F_i}), i = 4, 3, 2, \tag{6}$$

where $\mathrm{UP}(x, y)$ aims to upsample $x$ to the same scale of $y$.

After utilizing semantic compensation through the RCTVF module, we generate multi-scale semantic compensated visual features $Q^* = [Q_1^*, Q_2^*, Q_3^*, Q_4^*]$. Finally, $Q^*$ is fed into the segmentation head [22] to produce predictions $p \in \mathbb{R}^{C \times H \times W}$, which are compared with the labels or pseudo-labels to calculate the cross-entropy segmentation loss $L_{ce}$. The overall loss function for the training phase is expressed as follows, where $\lambda$ represents the training weight.

$$L = L_{orth} + L_{align} + \lambda L_{ce}. \tag{7}$$

Further details about our method and algorithm can be found in Appendix A.2.

# 4 Experiments

In this section, we first provide a detailed description of the experimental settings, including the datasets and implementation details, in §4.1. Subsequently, we present the main experimental results of the model in §4.2. Furthermore, in §4.3, we conduct comprehensive ablation studies to further validate the effectiveness of the CroPe.

## 4.1 Experimental Settings

**Datasets:** To demonstrate the effectiveness of our proposed CroPe method, we conduct experiments across all adverse scenes in seven real-world datasets, including Cityscapes (CS)[23], ACDC[24], Dark Zurich (DZ)[25], Nighttime Driving (ND)[26], BDD100K-Night (BD)[27], Foggy Zurich (FZ)[28] and Foggy Driving (FD) [29]. Detailed dataset information, including adverse scene types, data splits, and statistics, can be found in Appendix A.3.

**Implementation Details:** Following the prevailing method DAFormer, we adopt CLIP (-B/16 and -L/14 [30]) as the backbone. During training, we use a resolution of 512×512, rather than the high resolution of 1024×1024 employed by SOTA methods, and omit the FD loss typically used. The initial learning rate for the AdamW optimizer is set to 6e-5, and the learning rates for the encoder, RCTVF module, and segmentation head are 6e-5 scaled by $\frac{1}{10}\times$, $10\times$, $10\times$, respectively. Additionally, the context length of the text prompt $M$ is fixed to 5. The attention layers $N$ are set to 6, with two layers computed at each scale. The weight parameter $\lambda$ is set to 2.0. We conduct training experiments for 40,000 iterations. All modules are retained during inference.

## 4.2 Comparison with State-of-the-art Methods

Table 1 presents a comprehensive performance comparison between our CroPe and existing methods on seven datasets across ten challenging scenarios.

**Cityscapes to Foggy Scenes:** As shown in Table 1, we compare three foggy scenes, ACDC-Fog, FZ, and FD. Among them, we use CS as the source domain, ACDC-All or FZ as the target domain for training, and FD uses the model trained on FZ for direct generalization testing. CroPe achieves the SOTA performance on all three scenes. On the FD dataset, CroPe improves by 6.2 mIoU over DAEN and 4.2 mIoU over SAM-EDA. These improvements are probably attributed to CroPe's cross-modal semantic compensation strategy, which enhances the semantic density of visual features, enabling it to address the challenging conditions such as dense and light fog. It is worth noting that on the ACDC-Fog dataset, CroPe outperforms DAFormer by 10.7 mIoU and DAEN by 5.0 mIoU. These demonstrate that CroPe's cross-modal semantic compensation has effective adaptation and scalability abilities to this foggy scene.

**Cityscapes to Night Scenes:** The nighttime scenes pose the greatest low-visibility challenges. Our CroPe still outperforms nighttime-specific and scene-agnostic models on four nighttime scenes, as shown in Table 1. Specifically, CroPe achieves 62.3 mIoU on the CS → DZ, improving by 19.8

Table 1: Comparison of mIoU (%) across four adverse scenarios: Foggy, Night, Rainy, and Snowy. The best accuracy in each column is marked in bold, and the second highest is marked in underlined. '−' indicates experiments that cannot be implemented using a scene-specific models, or the results were not clear for the scene-agnostic models.

| Models | Foggy | | | Night | | | | Rainy | Snowy | All |
|---|---|---|---|---|---|---|---|---|---|---|
| | ACDC-Fog | FZ | FD | ACDC-Night | DZ | ND | BD | ACDC-Rain | ACDC-Snow | ACDC-All |
| **Scene-specific Models** | | | | | | | | | | |
| CuDA-Net [6] | 55.6 | 49.1 | 53.5 | - | - | - | - | - | - | - |
| FIFO [14] | - | 48.4 | 50.7 | - | - | - | - | - | - | - |
| FogAdapt [5] | - | 50.6 | 53.4 | - | - | - | - | - | - | - |
| SAM-EDA [31] | - | - | 56.4 | - | - | - | - | - | - | - |
| GCMA [25] | - | - | - | - | 42.0 | 45.6 | 33.2 | - | - | - |
| MCGDA [16] | - | - | - | - | 42.5 | 49.4 | 34.9 | - | - | - |
| SWG [3] | - | 51.3 | 54.2 | - | - | - | - | - | - | - |
| DAEN [32] | 65.6 | 54.2 | 54.0 | - | - | - | - | - | - | - |
| **Scene-Agnostic Models** | | | | | | | | | | |
| AdaptSeg [33] | - | 26.1 | 37.6 | - | 30.4 | 34.5 | 22.0 | - | - | - |
| DAFormer [7] | 48.9 | 40.8 | - | 44.7 | 48.5 | 51.8 | 33.9 | 59.9 | 53.7 | 55.4 |
| SePiCo [19] | 58.5 | - | - | 50.5 | 54.2 | 56.9 | 40.6 | 66.1 | 57.9 | 59.1 |
| STA [11] | 60.2 | 46.9 | 54.9 | 48.4 | - | - | - | 61.3 | 58.0 | 60.9 |
| HRDA [8] | 69.9 | 46.0 | - | 53.1 | 55.9 | - | - | 73.6 | 69.5 | 68.0 |
| MIC [9] | 67.0 | 53.3 | 56.6 | 57.2 | 60.2 | 58.6 | 41.3 | 72.3 | 66.6 | 70.4 |
| PASS [13] | **70.6** | 59.9 | 60.2 | 60.3 | 60.2 | 57.0 | 43.0 | **74.6** | 70.0 | 70.8 |
| **CroPe (Ours)** | **70.6** | **60.4** | **60.6** | **66.8** | **62.3** | **61.6** | **47.5** | 74.4 | **71.4** | **72.0** |

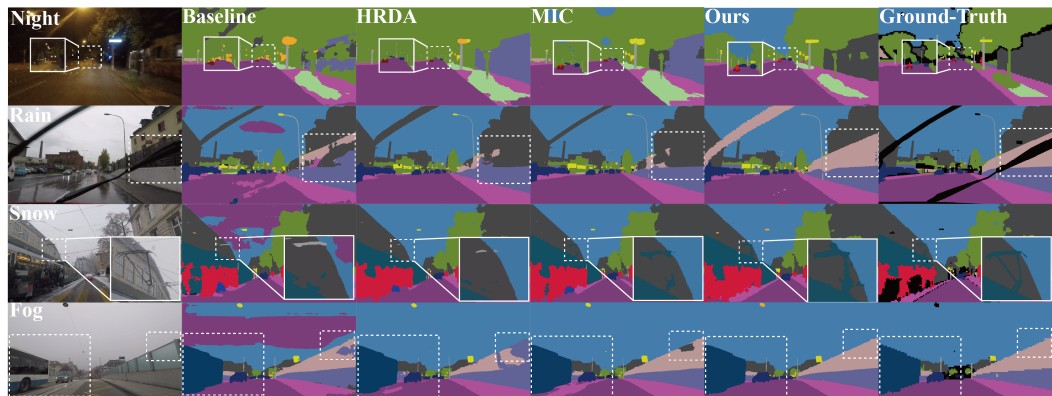

Figure 3: Visualization results comparison with SOTA methods across four adverse scenarios, white dashed boxes and zoomed-in boxes highlighting the different regions.

mIoU compared with MCGDA. In addition, the model trained on DZ shows excellent generalization ability when tested on the ND and BD datasets, improving by 4.6 mIoU and 4.5 mIoU over PASS, respectively. On ACDC-Night, CroPe also achieves the best performance, with 66.8 mIoU when trained on the CS → ACDC-All dataset and tested on ACDC-Night. These results indicate that CroPe is better at overcoming the ambiguity of category boundaries in night scenes.

**Comparison on Rainy, Snowy, and All Scenes:** As shown in Table 1, our CroPe trained on ACDC-All outperforms other methods in rainy, snowy, and all scenarios. On the ACDC-Rain dataset, CroPe achieved 74.4 mIoU, which is comparable to PASS. On the ACDC-Snow dataset, the CroPe even surpasses PASS by 1.4 mIoU, setting a new best benchmark. This performance showcases CroPe's ability to handle challenges such as blurry visual features and occlusions in rainy and snowy scenes while maintaining accurate modeling of both global semantics and small targets. On the ACDC-All dataset, CroPe achieved 72.0 mIoU, leading the SOTA method PASS by 1.2 mIoU. Finally, when the performance of ten datasets is averaged, CroPe surpasses PASS by 2.1 mIoU. The above results demonstrate the stable and comprehensive adaptability of CroPe.

**Visualization Results:** To clearly demonstrate the effectiveness of our method, we provide a visualization comparison with different methods across night, rain, snow, and fog scenes in Figure 3. As illustrated, the existing methods often produce hallucinations in adverse scenes, such as

Table 2: Ablation studies of proposed key components on CS→DZ.

| RCTVF | Prompt | CPTG | mIoU | gain |
|:-----:|:------:|:----:|:----:|:----:|
| ✗ | ✗ | ✗ | 59.6 | +0.0 |
| ✓ | ✗ | ✗ | 60.7 | +1.1 |
| ✓ | ✓ | ✗ | 61.1 | +1.5 |
| ✓ | ✓ | ✓ | 62.3 | +2.7 |

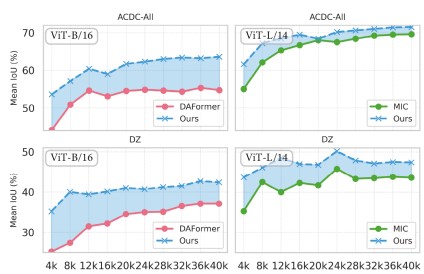

Figure 4: Convergence curve comparison.

Table 3: Internal ablation study of the CPTG module on the ACDC-All validation set.

| Inva | Spec | Comp | mIoU | gain |
|:----:|:----:|:----:|:----:|:----:|
| ✗ | ✗ | ✗ | 69.9 | - |
| ✓ | ✗ | ✗ | 69.6 | -0.3 |
| ✗ | ✓ | ✗ | 70.6 | +0.7 |
| ✓ | ✓ | ✗ | 70.4 | +0.5 |
| ✓ | ✓ | ✓ | 71.5 | +1.6 |

Table 4: The ablation study of the Reverse Decoding Chain in RCTVF. The "Mean" column represents the average mIoU on the ACDC-All and DZ validation sets.

| Method | ACDC-All | DZ | Mean |
|:-------|:--------:|:--:|:----:|
| No-Chain | 70.7 | 60.2 | 65.4 |
| Forward-Chain | 69.9 | 59.8 | 64.8 |
| Reverse-Chain | 72.0 | 62.3 | 67.1 |

misclassifying the sky as trees or blurring sidewalk boundaries, as emphasized by the rectangular box. In contrast, our CroPe generates more consistent representations by leveraging abstract semantic information from the text modality. Additional visualization results are in Appendices A.7 and A.8.

## 4.3 Ablation Studies

**Effectiveness of Individual Module.** Table 2 shows the ablation study of the key components of CroPe. The column "RCTVF" uses the RCTVF module and takes "a typical driving scenario with a [class]" fixed prompt and image as input, which improves 1.1 mIoU compared to pure vision methods. The column "Prompt" substitutes the invariant prompt with a learnable prompt, leading to an increase of 1.5 mIoU. The column "CPTG" leverages both the complementarity of invariance and domain awareness, achieving 62.3 mIoU. These results suggest that the proposed modules can produce synergistic effects, and additional ablation studies are detailed in the Appendix A.4.

Furthermore, Figure 4 illustrates the comparison of the convergence curve between Crope and the existing methods. Crope exhibits faster convergence speed, better stability, and higher accuracy.

**Component Analysis of the CPTG Module.** This section presents an ablation study of the CPTG module to assess the impact of each prompt, with results detailed in Table 3. The first row illustrates the performance of directly inputting $T$ into the RCTVF module. The column "Inva" applies Domain-Invariant Regularization to $T$, resulting in a decrease of 0.3 mIoU. This reduction occurs because it overlooks the coupling between the domain-invariant and domain-specific semantic structures in $T$, leading to blind constraints. The column "Spec" employs Domain-Specific Perception on $T$, effectively aligning the modalities and enabling $T$ to learn domain perception, which improves perfor-

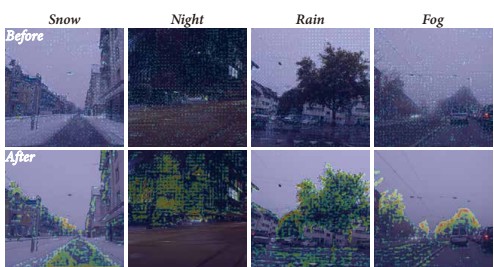

Figure 5: Visualization comparison of $F_1$ and $Q_1^*$ in the ACDC validation set.

mance by 0.5 mIoU. The column "Comp" introduces a decoupling strategy to address the blind constraint issue. Implementing Gated Complementary Fusion after decoupling enhances text embedding more effectively. Other ablation experiments of CPTG can be found in Appendix A.5.

**Component Analysis of the RCTVF Module.** This section presents a qualitative and quantitative analysis of the correctness and effectiveness of RCTVF. First, to confirm that multi-scale visual

Table 5: Ablation results of $L_{orth}$ and $L_{align}$.

| Methods | DZ | ACDC-All | ACDC-Night | ACDC-Fog | ACDC-Rain | ACDC-Snow |
|---|---|---|---|---|---|---|
| Crope | 48.7 | 71.5 | 54.2 | 78.2 | 72.1 | 73.0 |
| w/o $L_{orth}$ | 47.7 | 68.7 | 51.3 | 73.6 | 68.6 | 67.9 |
| w/o $L_{align}$ | 48.5 | 70.1 | 52.8 | 74.9 | 69.9 | 71.6 |

Table 6: Ablation results of CroPe on normal scene understanding tasks. For the training resolution, LR represents 512× 512, while HR represents 1024× 1024.

| Methods | Training Resolution | GTA5 to CS | SYNTHIA to CS |
|---|---|---|---|
| DAFormer | LR | 68.3 | 60.9 |
| MIC | LR + HR | 75.9 | 67.3 |
| CroPe | LR | 74.7 | 67.6 |

features can aggregate the semantics of $\widehat{T}$, we visualize the features $F_1$ and $Q_1^*$ in the first stage of the reverse chain text-visual fusion, as shown in Figure 5. Prior to RCTVF processing, the features of $F_1$ exhibited significant sparsity and window tracking issues. However, after fusing multi-level complementary-perceptive text embeddings and performing reverse decoding chain, the visual semantic information is effectively compensated and propagated (as shown in the second row of the figure). This proves that RCTVF achieves cross-modal semantic compensation as described in the method.

Secondly, the ablation experiment of the reverse decoding chain for transferring deep dense semantics is presented in Table 4. "No-Chain" indicates that only the Unified Attention of each layer is computed for semantic compensation, with no transmission between different scales. "Forward-Chain" refers to the transmission of semantics from the shallowest to the deepest layer. The performance of Forward-Chain is worse than No-Chain, with an average decrease of 0.6 mIoU. This is because the semantic information in the shallow layers lacks the quality and density necessary to enhance the expression of deeper features effectively. Additionally, when using the segmentation head to process visual features across multiple scales, shallow semantic information may fail to integrate effectively with deep features. In contrast, Reverse-Chain prioritizes the utilization of high-level features that carry rich semantic information, enhancing the overall semantic fusion effect. Additional ablation experiments related to RCTVF are detailed in Appendix A.6.

**Effectiveness of Loss Functions.** In our CroPe, two loss functions: $L_{orth}$ and $L_{align}$ are proposed. $L_{orth}$ serves as the core of the decoupling strategy, enabling the MLP to effectively decompose the text embedding $T$ into domain-invariant and domain-specific components. Removing this loss may cause the decoupling process to fail. Meanwhile, $L_{align}$ improves the generalizability of learnable prompts during training through generalizable hand-crafted templates (e.g., "a photo of a [class]"); removing $L_{align}$ would eliminate this crucial supervision signal. To verify the effectiveness of these two losses, we further conducted experiments to confirm the essential roles of $L_{orth}$ and $L_{align}$, as shown in Table 5. The results demonstrate that removing $L_{orth}$ leads to a 2.8 mIoU drop on the ACDC-All dataset, as MLPs fail to decouple, causing redundant and ineffective parameter learning. Likewise, removing $L_{align}$ eliminates the generalization constraint provided by hand-crafted prompts (e.g., "a photo of a [class]"), resulting in a 1.4 mIoU drop. These findings validate the critical role of our design.

**Effectiveness under Normal Scenes.** Although CroPe is primarily designed for adverse conditions such as fog or nighttime, it can also be applied to normal scene understanding. To verify this, we conducted experiments on two standard domain adaptation tasks (GTA5 → Cityscapes and SYNTHIA → Cityscapes). In these experiments, we adopted a general prompt "a photo of a [class]" instead of "a typical driving scenario with a [class]." As shown in Table 6, the results show that CroPe achieves improvements of 6.4 mIoU and 6.7 mIoU over the baseline (DAFormer), respectively. These findings are particularly encouraging, as they highlight the robust performance of CroPe even in scenarios with smaller domain gaps and less severe visual degradation, where the impact of our modules may be weakened. While CroPe demonstrates strong performance in adverse scenarios, these new

Table 7: Analysis of model complexity, training cost, and inference speed.

| Method | Params | Time | GPU Usage | Inference Speed | mIoU |
|---|---|---|---|---|---|
| HRDA(SegFormer) | 85.69 M | 17 h | 23.5 GB | 2.00 img/s | 68.0 |
| MIC(SegFormer) | 85.69 M | 23 h | 23.5 GB | 1.82 img/s | 70.4 |
| PASS(SegFormer) | 85.69 M | 25 h | 23.5 GB | 1.82 img/s | 70.8 |
| CroPe(ViT-B/16 w Frozen) | 27.36 M | 6 h | 5.7 GB | 5.12 img/s | 67.0 |
| CroPe(ViT-B/16 w Full) | 114.19 M | 9 h | 11.0 GB | 5.10 img/s | 68.6 |
| CroPe(ViT-B/16 w LoRA) | 43.91 M | 8 h | 7.2 GB | 4.98 img/s | 68.3 |
| CroPe(ViT-L/14 w Frozen) | 36.34 M | 9 h | 8.9 GB | 2.67 img/s | 69.7 |
| CroPe(ViT-L/14 w Full) | 341.37 M | 12 h | 18.0 GB | 2.64 img/s | 72.0 |
| CroPe(ViT-L/14 w LoRA) | 64.70 M | 10 h | 12.0 GB | 2.60 img/s | 71.4 |

Table 8: Comparison of the parameters in each module of the method.

| Method | Total | Visual Encoder | CPTG | RCTVF |
|---|---|---|---|---|
| MIC(SegFormer) | 85.69 M | 81.44 M (95.05%) | - | - |
| CroPe(ViT-L/14 w Full) | 341.37 M | 214.64 M (69.38%) | 10.24 M (2.99%) | 9.39 M (2.75%) |
| CroPe(ViT-L/14 w LoRA) | 64.70 M | 37.78 M (58.38%) | 10.24 M (15.83%) | 9.39 M (14.51%) |

experiments further validate the generalization ability and scalability of the proposed prompt-based adaptation framework in normal scenes.

**Complexity Comparison and Optimization.** Although the CroPe shows significant performance advantages, the increase in training parameters caused by its multi-modal design still raises concerns about deployment feasibility. This section quantitatively analyzes the efficiency of the CroPe through systematic experiments and introduces a lightweight adaptation strategy to optimize scalability. Table 7 shows the trainable parameters (Params), training time (Time), GPU memory usage (GPU Usage), inference speed, and mIoU performance comparison on the ACDC dataset. The experiment covers four types of models: 1) traditional SegFormer [34] variants; 2) CroPe variants with frozen backbones; 3) CroPe with full fine-tuning; 4) CroPe with LoRA [35] parameter-efficient fine-tuning.

We conduct all experiments on a single RTX4090. It can be seen that methods based on the SegFormer (lines 1-3) generally occupy nearly 24GB of memory, have an average training time of more than 20h, and an inference speed of less than 2 img/s. This is mainly due to its use of large-resolution training and multi-branch forward strategy to ensure performance. All forms of CroPe can better achieve the balance between efficiency and performance. For example, CroPe achieves certain performance with less than 10 GB of memory and less than 10 h of training time when the backbone network is frozen (lines 4 and 7). The fully fine-tuned CroPe further maximizes the performance (lines 5 and 8) without significantly increasing the training cost. However, the increase in its parameter raises a key question: Is CroPe still competitive under the same parameter adjustment? Therefore, we introduce the LoRA strategy to apply low-rank projections with rank $r = 64$, scaling factor $\alpha = 2r$, and dropout=0.1 to the $q$ and $v$ branches in ViT (lines 6 and 9), achieving a better performance-efficiency balance under limited limits. The specific parameter analysis of each module within CroPe are shown in Table 8.

## 5 Conclusion

In this paper, we present CroPe, a Cross-Modal Semantic Compensation Adaptation method for UDA scene understanding in adverse scenarios. We introduce the Complementary-Perceptive Text Generation module to enhance the cross-domain semantic representation of text and develop the Reverse Chain Text-Visual Fusion module to improve the consistency of multi-scale visual features by incorporating dense semantic embeddings of text. Extensive experiments demonstrate that CroPe enhances domain-invariant feature learning, alleviating model hallucinations and instability in various adverse scenes. However, CroPe has certain limitations, including an increase in model parameters. In future work, we will address these challenges and further optimize the model.

## Acknowledgements

We thank the anonymous referees for their constructive comments which have helped improve the paper. This work was supported by the National Natural Science Foundation of China (Grant Nos.: 62576006 and 72471001), the Natural Science Foundation for the Higher Education Institutions of Anhui Province (Grant No.: KJ2021A0038), the Open Research Fund of the State Key Laboratory of Brain-Machine Intelligence at Zhejiang University (Grant No.: BMI2400004), the Natural Science Foundation of Anhui Province (Grant No.: 2408085J037).

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

# A   Technical Appendices

The technical appendix and supplementary materials are organized as follows: 1) Section A.1 gives more related works to help quickly understand the state-of-the-art work in this field; 2) Section A.2 contains the architectural details and algorithmic supplements of CroPe; 3)Section A.3 contains detailed dataset information, including adverse scene types, data splits, and statistics. 4) Section A.4 provides additional complete component ablation studies; 5) Section A.5 performs an additional prompt ablation study in the CPTG module; 6) Section A.6 presents a quantitative analysis and additional experiments that examine the performance of the RCTVF module; 7) Section A.7 contains visualization of feature maps after processing by the RCTVF module; 8) Section A.8 contains more experiments on CroPe segmentation visualization; 9) Section A.9 contains potential societal impacts.

## A.1   More Related Work

In order to reduce the difference in feature distribution between domains, UDA proposes a variety of methods to bridge the gap between the source domain and the target domain. Existing research can be mainly divided into two paradigms: adversarial training and self-training. Adversarial training [36, 37, 38] aims to coordinate the outputs across domains by aligning the distributions of different domains at different levels such as input, features, and output. However, this method is often affected by instability [39], which limits its cross-domain effect. In contrast, self-training [40, 41] jointly optimizes the distribution of the source domain and the target domain through pseudo-label learning and gradually improves the performance of the target domain. In recent years, HRDA [8], which relies on self-training, achieves a balance between preservation of high-resolution detail and perception of long-range context through multi-resolution training (large low-resolution context + small high-resolution details) and a scale attention mechanism. The core challenge of this paradigm is how to extract reliable pseudo-labels. Therefore, many studies conduct extensive explorations from the perspectives of confidence threshold setting [42] and pseudo-label correction [43, 44]. Our work also belongs to the self-training paradigm and further proposes a better solution on this basis.

## A.2   Network Architecture Details

We propose a cross-modal semantic compensation method to improve the consistency of the model in adverse scenarios. CPTG enhances the domain awareness and generalization ability of text, while RCTVF compensates for visual semantics with improved textual clues.

CPTG decouples text embedding into two components (invariant and specific) through a decoupling strategy. It aligns the specific component with CLIP visual features via Domain-Specific Perception and regulates the generalization of the invariant component through Domain-Invariant Regularization. Gated Complementary Fusion then fully integrates the two decoupled features, providing rich text priors for RCTVF. In RCTVF, multi-scale visual features interact with text cues through Unified Attention, where visual features serve as queries and the embeddings generated by CPTG function as key-value pairs. This process effectively fuses multi-level textual semantics into the visual modality while providing semantic compensation. Additionally, we connect multi-scale features through a Reverse Decoding Chain to enhance the fusion effect and propagate refined semantics from fine-grained to coarse-grained layers. The complementary design of the two modules ensures that CPTG focuses on cross-domain guidance, while RCTVF tackles the issue of visual semantic sparsity. By applying segmentation loss to the multi-scale output of RCTVF, we achieve joint optimization of text and visual modalities, enabling simultaneous training for visual-text alignment and UDA adaptation performance. The complete training process of our CroPe is illustrated in Algorithm 1.

## A.3   Detailed Datasets Information

Cityscapes (CS) is captured under normal weather conditions in 50 European cities, containing 2,975 training images, 500 validation images, and 1,525 test images. ACDC contains four adverse scenes: fog, rain, snow, and night. For each scene, there are 400 training images, 100 validation images (including 106 nighttime images), and 500 test images. along with 1,600 clean reference images (ACDC-ref). Dark Zurich (DZ) provides 8,779 images captured during nighttime, twilight, and daytime, with 50 validation and 151 test images. Nighttime Driving (ND) includes 50 coarsely annotated nighttime images specifically designed for testing. BDD100K-Night (BD), a subset of the

---

**Algorithm 1** The core algorithm in CroPe

---

**Input:** Input image $I$, hand-crafted text prompt $m_1$ and learnable text prompt $m_2$, visual encoder $E_V$ and text encoder $E_T$.

**Output:** Multi-Scale Semantic Compensated Visual Features $Q^*$.

1: # Obtain encoder features for visual and textual modalities
2: Feed $I$ into $E_V$: $P = \{P_1, P_2, P_3, P_4\} = E_V(I)$.
3: Multi-scale visual features: $F = \text{FPN}(P) = \{F_1, F_2, F_3, F_4\}$.
4: Feed $m_1, m_2$ into $E_T$: $T_C = E_T(m_1), T = E_T(m_2)$.
5: # Complementary-Perceptive Text Generation (CPTG) module
6: Decouple the embedding $T$ into $T_I$ and $T_S$, and pass the orthogonal constraint function. {Eq. 1}
7: The soft consistency function constrains $T_I$ to stay close to $T_C$, maintaining the generality of $T_I$. {Eq. 3}
8: $T_S$ is combined with $\{P_2, P_3, P_4\}$ respectively to obtain the $T^* = \{T_1^*, T_2^*, T_2^*\}$. {Eq. 2}
9: $T_I$ is fused with $T^*$ to obtain multi-level complementary-perceptive text embeddings $\widehat{T} = [\widehat{T}_1, \widehat{T}_2, \widehat{T}_3]$. {Eq. 4}
10: # Reverse Chain Text-Visual Fusion (RCTVF) module
11: The projection $Q_{F_i}$ of $F_i$ and $\widehat{T}_{i-1}$ are processed by Unified Attention to get $Q_i^*$, which is then upsampled and fused with $Q_{F_{i-1}}$, $i = 4, 3, 2$ through Reverse Decoding Chain. {Eq. 5, Eq. 6}
12: The semantically compensated $Q^*$ is passed through the segmentation head to obtain logits $p$.

---

Table 9: The number of training, validation, and test sets for each dataset. "-" represents missing.

| Images | CS | ACDC-All | ACDC-Fog | ACDC-Night | ACDC-Snow | ACDC-Rain | DZ | ND | BD | FZ | FD |
|---|---|---|---|---|---|---|---|---|---|---|---|
| Traing set | 2975 | 1600 | 400 | 400 | 400 | 400 | 8779 | - | - | 3808 | - |
| Validation set | 500 | 406 | 100 | 106 | 100 | 100 | 50 | - | - | - | - |
| Test set | 1525 | 2000 | 500 | 500 | 500 | 500 | 151 | 50 | 87 | 40 | 101 |

BDD100K segmentation dataset, consists of 87 finely annotated nighttime images. Foggy Zurich (FZ) contains 3,808 images with light and medium fog, and 40 images for testing. Foggy Driving (FD) provides 101 annotated images purely for testing. For more structured statistics, see Table 9.

## A.4 Component Ablation Study

In the main text, we conducted an ablation study on the effectiveness of each module, focusing on the impact of incorporating the CPTG and RCTVF modules into our proposed CroPe model. Building on this, we provide a more comprehensive set of ablation experiments following the naming convention established in Table 2, including replacing the backbone network and choosing different CLIP variants to illustrate the evolution from the baseline DAFormer to our proposed CroPe.

The results are summarized in Table 10, where we added two columns for clarity: the column "V-only", which is checked if the visual backbone network of CLIP is used (ViT-B and ViT-L are two considered variants); and the column "w/o FD", which is checked if the FD loss strategy of DAFormer is discarded; otherwise, it is retained. We initially conducted experiments by replacing the backbone network alone (replacing SegFormer with CLIP's ViT-B or ViT-L). This modification leads to marginal performance gains, with mIoU improvements of 3.4 and 6.6, respectively. While replacing the backbone improves overall performance, it also exposes a key limitation: the FD loss in DAFormer (originally designed to align encoder features with those of a frozen encoder pre-trained on ImageNet) turns out to be suboptimal. This misalignment occurs because the CLIP encoder is not pre-trained on ImageNet, which can affect convergence efficiency and discriminability. To address this, we systematically remove the FD loss (indicated in the w/o FD column), resulting in further improvements of 3.4 and 4.5 mIoU for ViT-B and ViT-L, respectively. This validates the redundancy of the FD strategy when using a cross-modal backbone.

We then integrate handcrafted text prompts "a typical driving scenarios with a [class]" into the RCTVF module (column "RCTVF"), which improves the performance of ViT-B and ViT-L by 2.2 mIoU and 1.1 mIoU, respectively. This demonstrates that combining RCTVF with invariant information can enhance the semantic density of visual features through unified attention fusion and reverse decoding chain. Replacing the fixed text prompts with learnable prompts (column "Prompt") can capture

Table 10: Albation studies of proposed modules on CS→DZ using SegFormer, ViT-B/16 and -L/14.

| Method | Backbone | V-only | w/o FD | RCTVF | Prompt | CPTG | mIoU | gain |
|--------|----------|--------|--------|-------|--------|------|------|------|
| DAFormer | SegFormer | ✗ | ✗ | ✗ | ✗ | ✗ | 48.5 | - |
| CroPe | ViT-B/16 | ✓ | ✗ | ✗ | ✗ | ✗ | 51.9 | +3.4 |
| | | ✓ | ✓ | ✗ | ✗ | ✗ | 55.3 | +6.8 |
| | | ✓ | ✓ | ✓ | ✗ | ✗ | 57.5 | +9.0 |
| | | ✓ | ✓ | ✓ | ✓ | ✗ | 57.9 | +9.4 |
| | | ✓ | ✓ | ✓ | ✓ | ✓ | 59.4 | +10.9 |
| CroPe | ViT-L/14 | ✓ | ✗ | ✗ | ✗ | ✗ | 55.1 | +6.6 |
| | | ✓ | ✓ | ✗ | ✗ | ✗ | 59.6 | +11.1 |
| | | ✓ | ✓ | ✓ | ✗ | ✗ | 60.7 | +12.2 |
| | | ✓ | ✓ | ✓ | ✓ | ✗ | 61.1 | +12.6 |
| | | ✓ | ✓ | ✓ | ✓ | ✓ | 62.3 | +13.8 |

Table 11: Ablation experiments using different hand-crafted text prompts, the best results on each dataset are shown in bold.

| Source | ACDC Night | ACDC Snow | ACDC Rain | ACDC Foggy |
|--------|-----------|-----------|-----------|------------|
| | "a photo of a [class]" | | | |
| | 60.7 | 68.8 | 72.0 | 67.3 |
| | "a clean origami of a [class]" | | | |
| CS | 61.2 | 68.7 | 72.2 | 67.4 |
| | "an image of a driving with a [class]" | | | |
| | 59.7 | 67.4 | 72.0 | 67.0 |
| | "a typical driving scenario with a [class]" | | | |
| | **61.5** | **69.3** | **72.5** | **68.1** |

domain-specific features more flexibly, leading to an additional improvement of 0.4 mIoU. Finally, incorporating a complementary perceptual mechanism (column "CPTG") preserves both domain invariance and domain awareness, ultimately achieving mIoU scores of 59.4 and 62.3 for ViT-B and ViT-L, respectively. These findings highlight the efficacy and synergistic benefits of the proposed components, providing a strong rationale for the design of the CroPe model.

### A.5 More CPTG Ablation Studies

In the main text, we conducted ablation experiments inside the CPTG module to explore the effects of using a certain type of text prompt alone and in combination with different prompts. In this section, we further conduct an ablation study on the CPTG module to further evaluate the processing effects of various prompts in CPTG and give a visual analysis of CPTG.

First, we conducted selection experiments on the hand-crafted text prompts listed in Table 11. Considering that the knowledge based on ViT-B is more unstable and more sensitive to hand-crafted text prompts, we chose this model to verify the effects of various prompts in four different scenarios of the ACDC dataset. The experimental results show that the prompt "a typical driving scenarios with a [class]" shows significant advantages in all indicators. This is mainly due to the fact that the prompt provides a more general semantic representation of the driving scene, which is closer to the actual application scenario than "a photo of a [class]". However, the latter can still achieve relatively ideal results, and in future domain adaptation tasks involving non-driving scenarios, more general prompts obviously have greater application potential.

Secondly, through the qualitative ablation analysis of the CPTG in Figure 6, we can intuitively understand the necessity of the CPTG. The results show that in the absence of the CPTG, cross-layer fusion relying solely on visual features cannot effectively alleviate the serious visual information occlusion problem. By introducing semantic compensation of text prompts, the sparsity of features can be significantly improved, thereby effectively alleviating the serious domain shift problem.

### A.6 More RCTVF Ablation Studies

In the main text, we illustrated the impact of the RCTVF module's fusion strategies and the feature distribution before and after RCTVF processing through various figures and tables. To further

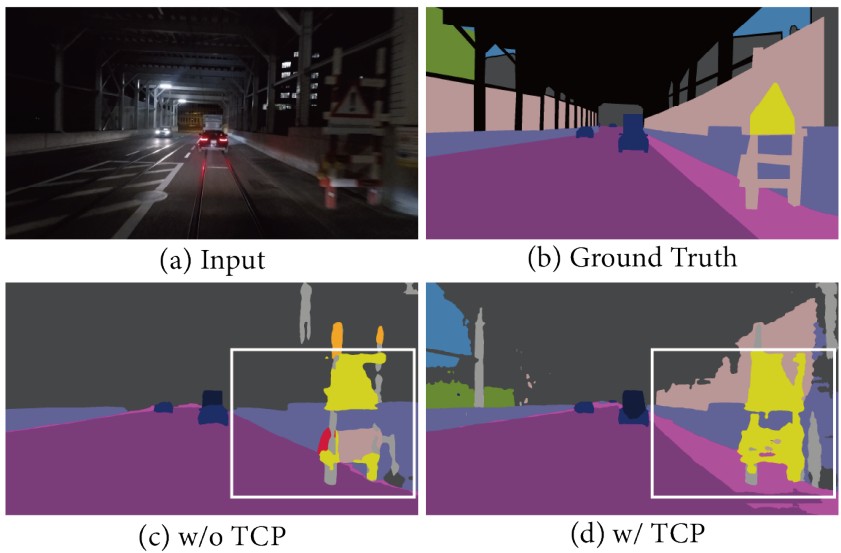

|  |  |
|:---:|:---:|
| (a) Input | (b) Ground Truth |
| (c) w/o TCP | (d) w/ TCP |

Figure 6: Use our CroPe model to perform visual comparison of the ACDC-All validation set samples with and without the CPTG module.

Table 12: RCTVF ablation on the ACDC-All validation set, where * denotes the visual modality only and "OOM" indicates out-of-memory.

| Batch Size | MIC | Ours* | Ours |
|:---:|:---:|:---:|:---:|
| 1 | 66.5 | 67.9 | 69.8 |
| 2 | 69.6 | OOM | 71.5 |

quantify the domain adaptation performance gains brought by the RCTVF module, we present the Maximum Mean Discrepancy (MMD) metric between the two domains for each category in the Cityscapes → ACDC-All task, as shown in Figure 7. The results demonstrate that CroPe achieves significantly lower MMD distances compared to MIC across all categories. Notably, in the more challenging categories such as train, rider, and motorcycle, CroPe further reduces the MMD by 0.0796, 0.1169, and 0.0500, respectively. This reduction highlights RCTVF's capability to extract more robust feature representations, effectively narrowing the domain gap and enabling the model to better learn domain-invariant features.

Furthermore, while the RCTVF module in the main text was presented solely within the context of a vision-text multimodal approach, it is also designed to handle pure visual modality inputs, resembling a self-attention mechanism. To explore this versatility, we compare the performance of RCTVF under pure visual modality and cross-modal strategies in Table 12. The results reveal that, although the pure visual modality (third column) outperforms MIC (second column)—underscoring RCTVF's adaptability and effectiveness—it remains constrained by inherent visual interference, which limits its ability to capture dense semantic information effectively. The introduction of the textual modality (fourth column) mitigates this limitation, further improving performance and reinforcing the necessity of cross-modal semantic compensation for effective domain adaptation. These findings collectively affirm the RCTVF module's critical role in enhancing robustness and semantic richness in UDA.

### A.7    Feature Visualization Experiment

To verify the effectiveness of semantic compensation of visual features across multiple scales as pointed out by the RCTVF module, we visualized the feature maps of $Q^* = \{Q_1^*, Q_2^*, Q_3^*, Q_4^*\}$ on the validation set of each scene of ACDC. As shown in Figure 8, the proposed CroPe demonstrates hierarchical semantic refinement and cross-scale consistency enhancement. Specifically, the deepest scale $Q_4^*$ absorbs dense textual semantics from $\widehat{T}_3$ and resolves the ambiguity caused by domain shift; while the reverse decoding gradually propagates high-level semantics to shallow scales, thereby sharpening the details of small objects (e.g., road poles in night scenes). This experiment further

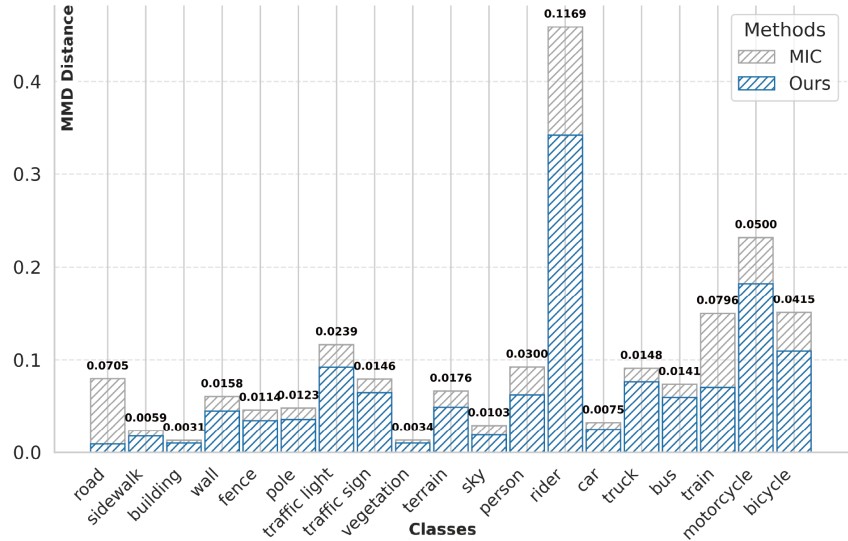

Figure 7: Comparison of Maximum Mean Discrepancy (MMD) distance, where the values represent the differences for each category.

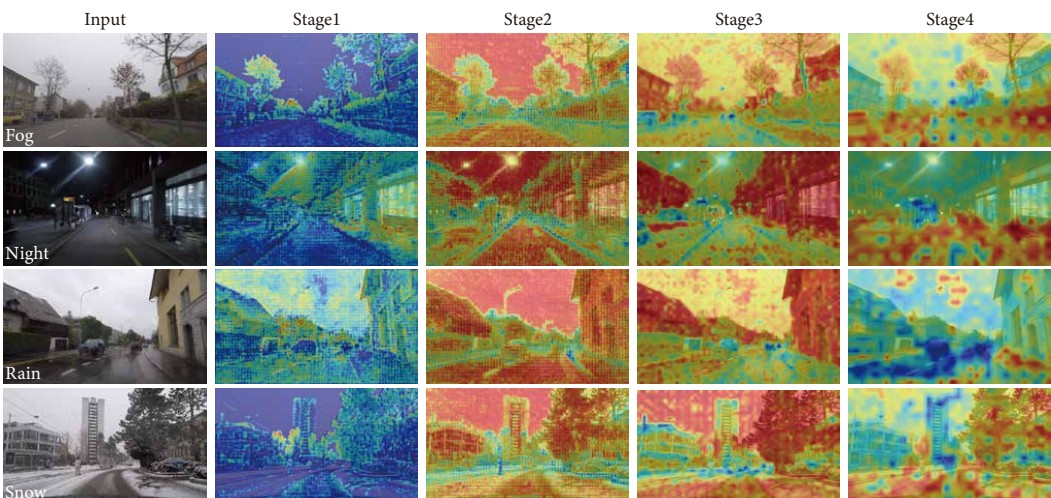

Figure 8: Visualization of the feature maps of the semantically compensated visual features of each stage (scale) of the CroPe model on the ACDC-Fog/Night/Rain/Snow validation set.

confirms that semantic compensation enhances feature density and domain-invariant representation learning, resulting in clearer semantic boundaries and fewer misclassifications in adverse scenes.

## A.8 More Segmentation Visualization Comparison

In Figures 9-12, we show qualitative segmentation results compared with Baseline (DAFormer) and MIC on the validation set of four different scenes: Cityscapes → ACDC-(Fog, Rain, Night, Snow). Compared with MIC, the masks predicted by our model have finer details near the object boundaries, thanks to CroPe's cross-modal semantic compensation, which makes up for the shortcomings of visual semantic sparsity. Compared with the Baseline, we have significantly reduced the hallucination phenomenon of the model. It should be noted that CroPe does not actually use DAFormer's FDloss and Segformer backbone, which we have explained in the text.

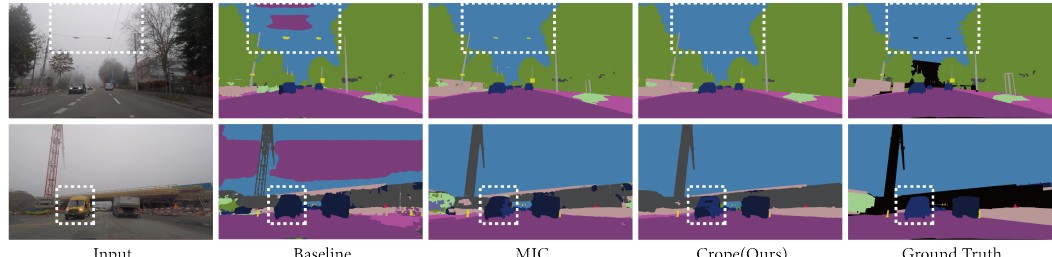

Figure 9: Qualitative visual comparison of the proposed CroPe with existing state-of-the-art methods on the ACDC-Fog validation set

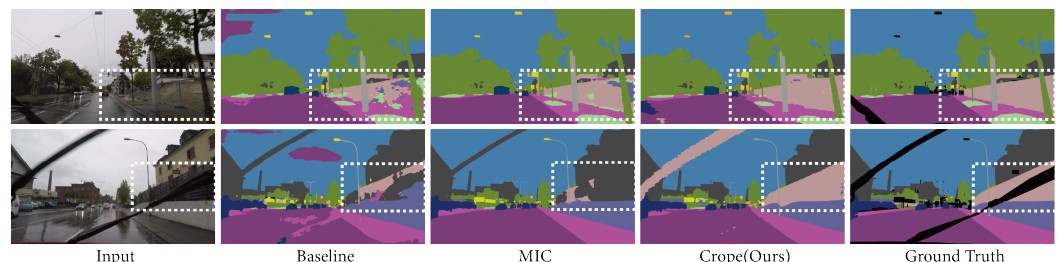

Figure 10: Qualitative visual comparison of the proposed CroPe with existing state-of-the-art methods on the ACDC-Rain validation set

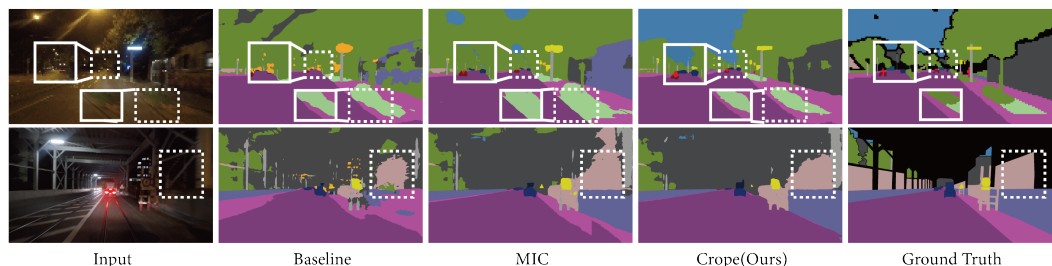

Figure 11: Qualitative visual comparison of the proposed CroPe with existing state-of-the-art methods on the ACDC-Night validation set

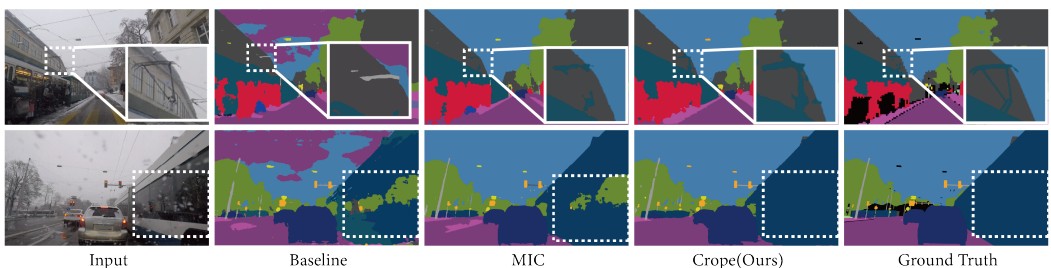

Figure 12: Qualitative visual comparison of the proposed CroPe with existing state-of-the-art methods on the ACDC-Snow validation set

## A.9 Potential Negative Societal Impacts

Our method poses no ethical risks regarding dataset usage or privacy violations, as all datasets and tools are publicly available and transparent.

