# OpenReview forum: "CroPe: Cross-Modal Semantic Compensation Adaptation for All Adverse Scene Understanding"
_NeurIPS.cc/2025/Conference — NeurIPS 2025 poster_

### Official Review · Reviewer_p1zo · 2025-06-22

**Clarity:** 3
**Significance:** 2
**Originality:** 2
**Rating:** 4
**Confidence:** 3

**Summary:**

This paper proposes a framework CroPe to incorporate text modality for scene understanding in adverse weathers. The model augments visual features with text through a multi-scale visual feature extractor together with multi-level text embeddings to extract features useful for semantic segmentation. The authors have conducted extensive experiments in various weather settings and with different training schemes to demonstrate the effectiveness of the framework.

**Questions:**

1. Can the authors provide more clarification on the term Cross-Modal Semantic Compensation? At inference time, how is the handcrafted text prompt being generated? Is the text prompt tied to fixed scene assumptions (e.g. driving) and how does this affect generalization?
2. The paper claims that previous methods tend to have hallucinations at several places. What are the intuitions behind this and are there any specific references that discuss this issue? Also how does the author quantitatively compare hallucination of the proposed model versus the baseline models?

**Ethical Concerns:**

["NO or VERY MINOR ethics concerns only"]

**Final Justification:**

The authors' rebuttals and responses have addressed some of my concerns. However, I think the paper overall would benefit from more editing to improve the clarity of the method, with clearer motivations for each introduced module (especially the design of text prompt related modules).

**Limitations:**

The authors mentioned one limitation of increased model parameters. More discussion can be included on potential generalization challenges (e.g., unseen categories or scene types) or the dependency on prompt engineering.

Suggestions:
1. Table and Figure captions in the experiment section can be improved by providing a concise summary of the main observations. For instance, what is the reader expected to see from the feature visualization in Figure. 5?
2. Given the moderate size (parameter count) of the model is around ~100 to 300M I think it is not particularly informative to include LoRA experiments as they are usually applied for much bigger pre-trained models. Most of Table 5. could be moved to Appendix.
3. Is there any ablation results on the effectiveness of the orthogonality constraint?

**Paper Formatting Concerns:**

/

**Quality:**

2

**Strengths And Weaknesses:**

Strengths:
1. The methodology figure (Fig. 2) is well-made and clearly demonstrated the key designs of the proposed framework.
2. Experiments contain comprehensive comparisons with various baseline models and under different adverse weather settings.

Weaknesses:
1. The main contribution has limited novelty. The model incorporates CLIP text embeddings with feature pyramid network (FPN) which is common for semantic segmentation and detection models since MaskRCNN. The main experiments is on how to incorporate those text embeddings into the network through common attention/fusion operations.
2. More explanations of the learnable and handcrafted text prompts are needed as they serve the core component of this model. For instance, the descriptions on handcrafted prompts can be moved to the main paper instead of in the appendix only.
3. The experiment section need to contain more in-depth discussion of why the proposed model outperforms similar prior work and does not need to re-iterate the quantitative performance gain as they are already clear in the table. The term 'hallucination' occurs frequently as weaknesses of existing baselines but is not quantitatively measured.

---

> ### Author Rebuttal · Authors · 2025-07-30
>
> Thanks for reviewing our work and providing these valuable suggestions and comments.
>
> **W1:** While our method leverages commonly known components such as CLIP text embeddings and FPN, we emphasize that their integration to compensate for severely degraded visual features (e.g., night, fog) is novel in unsupervised domain adaptive semantic segmentation (UDASS).
>
> First, to the best of our knowledge, our work is the first to apply CLIP text embeddings in unsupervised domain adaptive semantic segmentation (UDASS) to compensate for domain gap caused by severe visual degradation. In contrast, existing CLIP-based methods, such as WeakCLIP [1] and TP-SIS [2], typically generate segmentation predictions directly from vision-language score maps, facing challenges in UDASS due to insufficient visual features and limited prompt expressiveness. Our approach introduces a decoupled prompt design that separates domain-invariant and domain-specific text embeddings, coupled with a gated complementary fusion module to dynamically integrate domain-specific and domain-invariant representations. This design surpasses conventional attention/fusion operations, significantly enhancing the adaptability of CroPe to complex domains.
>
> Second, our method achieves a 6.5 mIoU improvement on the CS to ACDC-Night dataset (Table 1) while also improving training and inference efficiency (Table 5). These results demonstrate the practical value and effectiveness of our proposed design.
>
> In summary, our unique perspective, task-specific design, and substantial performance gains represent a significant and non-trivial contribution. We will emphasize these points more clearly in the revised manuscript.
>
> [1]WeakCLIP: Adapting CLIP for Weakly-Supervised Semantic Segmentation, IJCV 2025.
> [2]Text Promptable Surgical Instrument Segmentation with Vision-Language Models, NeurIPS 2023.
>
> **W2:** As described in the paper (Page 3), the learnable prompt is integrated into the CPTG and RCTVF modules, providing semantic compensation for severely degraded visual features and serving as the core prompt of the model. The handcrafted prompt acts as an auxiliary supervision signal during training, enhancing the generalization of the learnable prompt and preventing overfitting to source domain scenarios. Due to page constraints in the initial submission, some details are placed in the appendix. In the revised manuscript, we will integrate these critical details into the main text to highlight their central role. Thank you again for your insightful suggestions.
>
>
> **W3:** The experiment section is described to provide clear and objective final performance metrics, enabling readers to quickly understand the performance of CroPe. Regarding the depth analysis of the reasons behind performance improvements, we have primarily presented these through the ablation study. For instance, switching to the CLIP encoder leverages its larger pre-trained knowledge to enhance expressive capacity (Table 8); discarding ImageNet feature distance regularization allows the model to focus more on specific scenes, as the CLIP encoder does not rely on retaining ImageNet pre-trained knowledge (Table 8); the RCTVF module, through semantic compensation, outperforms existing vision models even with pure visual input (Table 10); and CroPe achieves higher training efficiency with smaller resolutions and fewer computational strategies (Table 5). To further improve, we will provide an in-depth analysis in the revised manuscript on how these designs address the extreme visual degradation challenges in UDASS.
>
> Regarding the hallucination issue, mIoU or MMD (Figure 7) metrics can be used for quantification. A detailed discussion is provided in **Q2**.
>
> **Q1:** Below is our clarification for all questions.
>
> 1) **Term explanation:** Cross-modal semantic compensation refers to integrating the high-density semantics of the text modality into the visual modality to eliminate visual feature degradation in adverse scenarios (e.g., fog, night). It is well known that the visual features are sensitive to changes in illumination, texture, and occlusion patterns, while the texts providing a class inforamtion can ensure the semantic stability across different conditions and domains. The stable textual features offer semantic compensation for visual features in cross-domain settings, thereby improving the generalization ability of cross-domain sementic segmentation.
>
> 2) **Explanation of handcrafted prompts at inference time:** Handcrafted prompts are not required at inference time, we only use learnable prompts. The handcrafted prompts (e.g., “a photo of a [class]”) are only used during training as generalization constraints to regularize the learning process and prevent the learnable prompts from overfitting to the source domain (e.g., Cityscapes), thereby improving generalization to target domains.
>
> 3) **Fixed scene assumptions:** Text prompts are not inherently tied to scene assumptions, and they do not affect generalization. Specifically, in Table 9 of our paper, we compare multiple handcrafted prompts, including generic "a photo of a [class]" and the more scene-specific "a typical driving scenario with a [class]". The two prompts differ by only a marginal 0.8 mIoU compared to our overall improvement of 6.5 mIoU in ACDC-Night dataset, indicating that the generic prompt is also sufficient for achieving strong performance, without affecting generalization. We use the scene-specific prompt "a typical driving scenario with a [class]" because current UDA benchmarks, especially those under adverse conditions, are almost exclusively focused on driving scenes (e.g., TPAMI 2023: Domain adaptive and generalizable network architectures and training strategies for semantic image segmentation). Hence, prompts like "a typical driving scenario..." are designed specifically to enhance performance under these conditions. For broader applicability, prompts such as "a photo of a [class]" offer more general semantic coverage and can be used for non-driving domains.
>
> 4) **Prompt generalization:** We conduct experiments on GTA5-to-Cityscapes and SYNTHIA-to-Cityscapes datasets, both of which represent normal scenes rather than adverse scenes, to evaluate the generalization ability of the “a photo of a [class]”.
> | Methods  |    Training Resolution    | GTA5 to Cityscpes | SYNTHIA to Cityscapes |
> |:--------:|:-------------------------:|:-----------------:|:---------------------:|
> | DAFormer |        LR(512×512)        |       68.3        |         60.9          |
> |  CroPe   |        LR(512×512)        |       74.7        |         67.6          |
>
> The results show that CroPe achieves improvements of 6.4 mIoU and 6.7 mIoU over the baseline (DAFormer), respectively. These results highlight robust performance of CroPe, even in scenarios with smaller domain gaps and milder visual degradation. The prompt "a photo of a [class]" also maintains good generalization.
>
> **Q2:**  Below is our clarification for all questions.
>
> 1) In this paper, hallucinations refer to models erroneously predicting non-existent environmental elements based on prior knowledge. For example, due to the high joint probability between “road” and “sidewalk,” a model may incorrectly predict sidewalks alongside a “road” where none exist. In adverse UDASS scenarios, such hallucinations are more severe. Adapting to multiple adverse conditions (e.g., ACDC-Fog, ACDC-Night) requires learning joint distributions for each source-target domain pair, yet knowledge from foggy conditions (e.g., ACDC-Fog) often proves unreliable in nighttime settings (e.g., ACDC-Night), leading to mispredictions. ProMaC [1] has clarified this hallucination problem, while MCGDA [2] and Fifo [3] further note that significant texture and spatial structure variations in nighttime and foggy conditions aggravate erroneous predictions.
>
> 2) The hallucination can be quantified by UDA performance (e.g., mIoU) or by assessing enhanced class distinguishability using the MMD metric (Figure 7).
>
> [1] Leveraging Hallucinations to Reduce Manual Prompt Dependency in Promptable Segmentation, NeurIPS 2024.
> [2]Map-guided curriculum domain adaptation and uncertainty-aware evaluation for semantic nighttime image segmentation, TPAMI 2020.
> [3] Fifo: Learning fog-invariant features for foggy scene segmentation, CVPR 2022.
>
> **Limitations:** Thank you for your valuable suggestions. We will discuss further potential challenges with generalization or limitations of the model in the revised manuscript.
>
> **S1:** We will revise the captions of figures and tables to summarize key observations concisely.
>
> **S2:** In the revised manuscript, we will move Table 5 to the Appendix and include additional analysis of key modules to emphasize core experimental results.
>
> **S3:** The orthogonal loss $L_{orth}$ is a pivotal component of our decoupling strategy. When mapping text embeddings $T$ with MLPs, $L_{orth}$ is essential by default, as its absence prevents effective decoupling. Thus, no separate ablation was needed. To address your concern, we conducted additional experiments confirming the critical role of $L_{orth}$ in decoupling.
> | Methods | CS to DZ | CS to ACDC-All | CS to ACDC-Night | CS to ACDC-Fog | CS to ACDC-Rain | CS to ACDC-Snow |
> |:--:|:--:|:--:|:--:|:--:|:--:|:--:|
> | CroPe |  48.7|  71.5| 54.2 | 78.2 | 72.1 | 73.0 |
> | w/o $L_{orth}$ | 47.7 | 68.7 | 51.3 | 73.6 | 68.6 | 67.9 |
>
> The results show that removing $L_{orth}$ leads to a 2.8 mIoU drop on the ACDC-All dataset, as MLPs fail to achieve decoupling. We will incorporate these analyses into the revised manuscript.

---

> > ### Author Response · Authors · 2025-08-04
> >
> > Dear Reviewer p1zo,
> >
> > We would like to express our gratitude to you for your time and effort in reviewing our manuscript and especially providing constructive
> > comments and valuable suggestions for our manuscript. We have provided detailed responses to the concerns raised.
> >
> > We would greatly appreciate your feedback on whether we have fully addressed your concerns. Your insights are valuable to us and will help improve the quality of our paper.
> >
> > Thank you once again for your time and consideration.
> >
> > Best regards,
> >
> > The authors of CroPe

---

> ### Comment · Reviewer_p1zo · 2025-08-05
>
> Thanks for the additional clarifications and quantitative results.
> I still have some questions related to previous discussion:
>
> - Q1-1: For the part *"It is well known that the visual features are sensitive to changes in illumination, texture, and occlusion patterns..."* I wonder if this claim has any reference? My experience is that lower-level features are more aware of illumination changes, but visual features learned at deeper layers of neural nets can also incorporate more high-level, semantic features that are more robust to those changes.
>
> - Q1-2: what are the classes used by text prompts?
>
> - From other reviewers' discussions, the authors mentioned that handcrafted prompts are designed to "*constrain learnable prompts during training to ensure generalization, as they tend to overfit the source domain (labeled) rather than the target domain (unlabeled) in the task setting.*" However, the authors also mentioned that text modality are expected to be more stable and the contributing factor for domain-invariant regularization. Are those two claims conflicting?

---

> ### Author Response · Authors · 2025-08-05
>
> Thank you for your valuable comments.
>
> **Q1-1:** We agree that high-level features exhibit robustness in normal scenarios, but under adverse conditions, such as illumination changes, texture variations, or occlusion, high-level semantic features probably suffer from significant degradation. Several studies have claimed the sensitivity of visual features to adverse conditions. For instance, PASS [1] reports that "previous (visual-based) methods excessively focus on weather-specific features during the adaptation process, resulting in models overly sensitive to these specific visual features and lacking generalization across all adverse scenes". DTP [2] states that "in night-time scenes, the entanglement of content and complicated lighting lead to confused semantics", which means that semantic features are not robust to adverse conditions. In addition, Rainy WCity [3] claims that "the performance of these (visual-based) methods drops sharply when facing inclement weather, since adverse weather can significantly degrade the image quality and readability".
>
> References:
>
> [1] Parsing all adverse scenes: Severity-aware semantic segmentation with mask-enhanced cross-domain consistency, AAAI 2024.
>
> [2] Disentangle then parse: Night-time semantic segmentation with illumination disentanglement, ICCV 2023.
>
> [3] Rainy WCity: A Real Rainfall Dataset with Diverse Conditions for Semantic Driving Scene Understanding, IJCAI 2022.
>
> **Q1-2:**  The classes used by text prompts are the labels of each class, such as 19 categories: [road, sidewalk, building, wall, fence, pole, traffic light, traffic sign, vegetation, terrain, sky, person, rider, car, truck, bus, train, motorcycle, bicycle]. The category labels of all classes are relevant to the task. The design process for the text prompts is explained in lines 104 to 130 of our paper.
>
> **Q3**：We believe that the two claims are not conflicting. Our statement that "the text modality is expected to be more stable" refers to its ability to provide consistent, domain-invariant category semantics (e.g., "a photo of a [class]"). With this in mind, we introduce two types of text prompts,  which are expected to provide more stable ability. Among the two prompts, the learnable prompts offer greater flexibility, they can be dynamically adapted to specific scenarios, providing enhanced representation ability, but are influenced by specific source-target domain pairs during training and are prone to overfitting the source domain, which limits their generalization to the target domain. Thus, we also employ handcrafted prompts as a form of domain-invariant regularization to constrain the optimization of learnable prompts, thereby enhancing the generalizability of learnable prompts. This approach leverages the adavantages of handcrafted prompts and learnable prompts, as described in lines 128–130 of our paper.

---

> > ### Comment · Reviewer_p1zo · 2025-08-06
> >
> > Thanks the authors for the detailed responses.
> > - For Q1-1 since all the citations are for adverse weather / nighttime scenes I recommend rephrasing the original sentence in the paper to indicate this setting and with proper citations to make it more rigorous.
> > - For Q1-2 it would be helpful to add those classes in the appendix to supplement the main paper.
> >
> > Since the authors' rebuttals have addressed some of my concerns I will increase the rating to 4.

---

> > > ### Author Response · Authors · 2025-08-07
> > >
> > > We sincerely thank you for all your valuable comments which enable us to further improve our paper.
> > >
> > > - For Q1-1, we will rephrase the original sentence to explicitly indicate the setting of adverse weather and nighttime scenes, and we will ensure proper citations are included to support this context.
> > >
> > > - For Q1-2, we will add the relevant classes in the appendix to provide supplementary information as recommended.
> > >
> > > We sincerely appreciate you raising the rating to 4. Your insights have greatly contributed to refining our paper.
> > >
> > > Thank you once again for your time and effort.

---

### Official Review · Reviewer_jRMA · 2025-06-30

**Clarity:** 2
**Significance:** 3
**Originality:** 3
**Rating:** 4
**Confidence:** 3

**Summary:**

The paper introduces CroPe, a novel Cross-Modal Semantic Compensation Adaptation framework designed for unsupervised domain adaptation (UDA) in adverse scene understanding tasks (e.g., fog, snow, rain, night). Unlike previous visual-only approaches, CroPe incorporates textual semantics via a vision-language framework to compensate for degraded visual features under challenging conditions.

**Questions:**

How robust is the model to poor or noisy prompts (e.g., misphrased or low-quality language inputs)?

Does the proposed method also work well in normal scene understanding?

**Ethical Concerns:**

["NO or VERY MINOR ethics concerns only"]

**Final Justification:**

Considering some minor concerns about efficiency and visualizations remain unaddressed, I tend to maintain my score at this stage.

**Limitations:**

The full fine-tuning version increases parameters significantly, posing challenges for deployment on resource-constrained systems.

The proposed method still depends on manually designed prompts; automated or adaptive prompt generation remains unexplored.

They assume paired vision-text representation (e.g., CLIP) and may struggle in environments with weak textual priors or unseen categories.

**Paper Formatting Concerns:**

Figure 1, 4, 6, and 7 are in a pretty low resolution and poor quality.

**Quality:**

3

**Strengths And Weaknesses:**

Weaknesses:
1. Domain-specific and domain-invariant representation learning is a traditional and popular approach in addressing domain shift problems. Though the author investigates the literature of UDA, they should better clarify the difference between the proposed method and existing methods, such as methods in [1].

[1] Wang J, Lan C, Liu C, et al. Generalizing to unseen domains: A survey on domain generalization[J]. IEEE transactions on knowledge and data engineering, 2022, 35(8): 8052-8072.
[2] Zhou K, Liu Z, Qiao Y, et al. Domain generalization: A survey[J]. IEEE transactions on pattern analysis and machine intelligence, 2022, 45(4): 4396-4415.


2. Incorporating text may introduce architectural and training complexity compared to vision-only baselines, and heavily rely on manual prompt.

3. CroPe(ViT-L/14 w Full) has 341.37M, which largely impedes the application in real-world scenarios. Please clarify this.
4. Performance relies on handcrafted textual prompts (e.g., "a typical driving scenario with a [class]"), requiring careful design per task/domain.

---

> ### Author Rebuttal · Authors · 2025-07-30
>
> Many thanks for reviewing our work and providing these valuable suggestions and feedback.
>
> **W1:** Thanks for reviewer's suggestion. Existing domain-specific and domain-invariant representation learning methods in [1] and [2], such as domain adversarial learning [3], explicit feature alignment [4], invariant risk minimization [5], and generative modeling [6], primarily rely on the visual modality to align or disentangle features across domains. In contrast, our method innovatively decouples the internal embeddings of learnable text prompts, with domain-invariant embeddings capturing generalization and domain-specific embeddings enabling modality alignment by perceiving visual domain information. Furthermore, we introduce a gated complementary fusion module to dynamically balance the domain-perception and generalization capabilities of prompts. This fusion mechanism emphasizes a unique contribution, distinctly setting our work apart from prior methods. We will include a detailed comparison with related methods in the revised manuscript to highlight these differences.
>
> [1]Generalizing to unseen domains: A survey on domain generalization, TKDE 2022.
> [2]Domain generalization: A survey, TPAMI 2022.
> [3]Unsupervised Domain Adaptation by Backpropagation, ICML 2015.
> [4]Domain Adaptation via Transfer Component Analysis, TNNLS 2010.
> [5]Representation Learning via Invariant Causal Mechanisms, arXiv.
> [6]DIVA: Domain Invariant Variational Autoencoders, MIDL 2020.
>
> **W2:** Below is our clarification for all questions.
>
> 1) **Regarding architectural and training complexity:** CroPe does not significantly increase computational overhead. On the contrary, it achieves superior efficiency compared to vision-only methods through efficient module design and low-resolution training. Specifically, CroPe is trained at a 512×512 resolution, substantially reducing computational costs compared to traditional vision-based methods (e.g., MIC, PASS), which rely on high-resolution training and complex strategies (e.g., mask reconstruction or feature distance regularization). As shown in Table 5, CroPe reduces training time by 52%–64%, lowers memory usage by 28%–64%, and improves inference speed by approximately 45%. These results demonstrate that CroPe not only avoids introducing significant complexity but also enhances efficiency through a streamlined training pipeline, outperforming vision-based methods that depend on high-resolution training and additional forward-propagation strategies.
>
> 2) **Regarding manual prompts and generalization:** Manual prompts are not inherently tied to specific task/domain, and they do not affect generalization.
> First, we clarify that manual prompts serve as auxiliary supervision signals, constraining learnable prompts during training to ensure generalization, as they tend to overfit the source domain (labeled) rather than the target domain (unlabeled) in the task setting. Manual prompts are not used during inference, thus avoiding heavy reliance.
> Second, in Table 9 of our paper, we compare multiple handcrafted prompts, including generic "a photo of a [class]" and the more scene-specific "a typical driving scenario with a [class]". The two prompts differ by only a marginal 0.8 mIoU compared to our overall improvement of 6.5 mIoU in ACDC-Night dataset, indicating that the generic prompt is also sufficient for achieving strong performance, without affecting generalization. We use the scene-specific prompt "a typical driving scenario with a [class]" because current UDA benchmarks, especially those under adverse conditions, are almost exclusively focused on driving scenes (e.g., TPAMI 2023: Domain adaptive and generalizable network architectures and training strategies for semantic image segmentation). Hence, prompts like "a typical driving scenario..." are designed specifically to enhance performance under these conditions. For broader applicability, prompts such as "a photo of a [class]" offer more general semantic coverage and can be used for non-driving domains.
>
>
> **W3:** CroPe (ViT-L/14 w/ Full) comprises 341.37M parameters, with the CLIP-ViT-L/14 backbone accounting for the majority (214.64 M). To address resource-constrained real-world scenarios, we provide a lightweight variant CroPe(ViT-L/14 w LoRA) with only 64.70M parameters, lower than 85.69M of MIC, while maintaining excellent performance (71.4 mIoU). As shown in Table 5, this LoRA-based variant also offers significant efficiency benefits. It reduces training time by 56% (10 hours vs 23 hours of MIC), lowers GPU memory usage by 49% (12.0 GB vs 23.5 GB), and improves inference speed by 43% (2.60 images per second vs 1.82 for MIC). These results demonstrate that CroPe(ViT-L/14 w Full) is suitable for high-precision scenarios, while the LoRA variant offers an efficient, low-resource alternative for applications requiring high real-time performance, effectively balancing performance and practicality.
>
> **W4:**  See response of **Regarding manual prompts and generalization:** in **W2**.
>
> **Q1:** Intuitively, incorrect fixed prompts may impose misguided constraints on learnable prompts. To validate this, we conducted experiments comparing three settings: incorrect prompts, no prompts, and correct prompts.
> | Prompts | ACDC-All | ACDC-Night | ACDC-Fog | ACDC-Rain | ACDC-Snow |
> |-----------------------|:----------:|:------------:|:----------:|:-----------:|:-----------:|
> | a non-driving scenario with [class] | 69.1     | 50.6       | 75.3     | 69.7    | 70.8      |
> | no prompts  | 70.1     | 52.8       | 74.9     | 69.9      | 71.6      |
> | a typical driving scenario with a [class] | 71.5     | 54.2       | 78.2     | 72.1      |73.0|
>
> The results show that incorrect prompts yield the lowest average performance, consistent with our expectations. Incorrect prompts introduce erroneous semantic priors, constraining text embeddings in unintended directions. Notably, the no-prompt setting outperforms incorrect prompts in most cases. This indicates that correctly designed prompts enable robust adaptation, even in diverse ACDC-All scenarios.
>
> **Q2:** We conduct experiments on two normal scene understanding tasks (GTA5 to Cityscapes and SYNTHIA to Cityscapes), and the relevant results are shown below. Specifically, we conducted experiments using the generic prompt “a photo of a [class]” rather than “a typical driving scenario with a [class].”
> | Methods  | Training Resolution | GTA5 to Cityscpes | SYNTHIA to Cityscapes |
> |:--------:|:--:|:-----------------:|:---------------------:|
> | DAFormer | LR(512×512) |       68.3        |         60.9          |
> |   MIC    | LR(512×512)+HR(1024×1024) |       75.9        |         67.3          |
> |  CroPe   | LR(512×512) |           74.7        |          67.6             |
>
> The results show that CroPe achieves improvements of 6.4 mIoU and 6.7 mIoU over the baseline (DAFormer), respectively. These findings are particularly encouraging, as they highlight the robust performance of CroPe even in scenarios with smaller domain gaps and less severe visual degradation, where the impact of our modules may be weakened. While CroPe demonstrates strong performance in adverse scenarios, these new experiments further validate the potential and scalability of the prompts in other scenarios.
>
> **Limitations:** Thank you for pointing out these comments.
> 1) Please see **W3** for parameter details.
> 2) Please see **W2** for manual prompts details.
> 3) Regarding limitations in unknown environments: In Unsupervised Domain Adaptation (UDA) tasks, model transfer performance is typically evaluated using the intersection of source and target domain category sets. CLIP currently exhibits strong expressive capability for these categories, meeting task requirements. To further clarify, we will discuss more potential limitations in the revised manuscript.
>
> **Paper Formatting Concerns:** Thank you for pointing out the formatting issue. Upon careful review, we found that the image resolution was reduced during the conversion process, resulting in unclear visual quality. We have re-exported high-resolution versions of the affected figures and will update them in the revised manuscript to ensure clarity.

---

> > ### Comment · Reviewer_jRMA · 2025-08-05
> >
> > Thank the author for their valuable efforts on the response. They addressed most of my concerns. There remains some confusion:
> >
> > W2: Why does the proposed method only use the lower resolution?
> >
> > Q1: As incorrect fixed prompts yield the lowest average performance, the proposed method may be very vulnerable to out-of-distribution cases when the given prompt does not match well with the test data.
> >
> > Q2: It would be better to strengthen the performance of the proposed method in normal cases because normal scenarios should be more common than adverse ones.
> >
> > I strongly suggest to improve your visualization and insert these new results in the revised version, making the presentation clearer. At the current stage, I tend to maintain my score.

---

> ### Author Response · Authors · 2025-08-05
>
> Thank you for your valuable comments.
>
> **W2:** SePiCo [1] uses low-resolution (LR) inputs for training, yielding suboptimal results due to limited generalization ability. Most of the existing methods rely on high-resolution (HR) for training to pursue superior performance, significantly increasing computational complexity, while their performance drops notably in lower resolution settings (e.g., MIC achieves only 54.6 mIoU on Cityscapes-to-Dark Zurich with LR, versus 60.2 mIoU with HR). Observing these, to demonstrate the robustness and high scalability of our method, we adopt the low-resolution (LR) for training and compare with state-of-the-art high-resolution training models. CroPe outperforms the state-of-the-art HR-trained models (e.g., 62.3 mIoU on Cityscapes-to-Dark Zurich). This verifies the superiority of CroPe in leveraging LR inputs without sacrificing performance. We can elaborate on these advantages and potential extensions in the revised manuscript.
>
> References:
>
> [1]Sepico: Semantic-guided pixel contrast for domain adaptive semantic segmentation, TPAMI 2023.
>
> **Q1:** We sincerely appreciate your valuable comment. We assume that significant mismatches between handcrafted prompts and test data distributions are unlikely in practice. Nevertheless, we acknowledge this as a potential limitation and will further clarify the out-of-distribution robustness of our prompt design and mitigation strategies in the revised manuscript.
>
> **Q2:** We sincerely thank you for your valuable suggestion. Our paper focuses on adaptability in various adverse scenarios because many methods achieve good performance in normal scenarios, but their performance in adverse scenarios is generally insufficient. Therefore, we focus on the adverse scenarios which are of great practical value ([1-3]). In addition, we will supplement the performance in normal scenarios in the revised manuscript.
>
> References:
>
> [1] Parsing all adverse scenes: Severity-aware semantic segmentation with mask-enhanced cross-domain consistency, AAAI 2024.
>
> [2] Energy-Based Domain Adaptation Without Intermediate Domain Dataset for Foggy Scene Segmentation, TIP 2024.
>
> [3] Dannet: A one-stage domain adaptation network for unsupervised nighttime semantic segmentation, CVPR 2021.
>
> **Suggestion:** Thank you for your valuable suggestion. We must provide the improved visualization and insert the new results in the revised version, to make the presentation clearer.

---

### Official Review · Reviewer_U3Dv · 2025-06-30

**Clarity:** 3
**Significance:** 2
**Originality:** 2
**Rating:** 3
**Confidence:** 3

**Summary:**

This paper proposes a cross-modal semantic compensation adaptation method, CroPe, which generates multi-level text embeddings integrating generalization and domain awareness through a Complementary Perceptive Text Generation (CPTG) module. It also utilizes a Reverse Chain Text-Visual Fusion (RCTVF) module to sequentially fuse compensation information from deep to shallow features, maximizing compensation gain to address the challenge of visual appearance degradation in scene understanding.

**Questions:**

See Weaknesses.

**Ethical Concerns:**

["NO or VERY MINOR ethics concerns only"]

**Final Justification:**

This framework is an engineering work that focuses on practical problems, but it lacks novelty and innovation. I tend to maintain the original score.

**Limitations:**

yes.

**Quality:**

3

**Strengths And Weaknesses:**

### **Strengths**:
1. CroPe introduces the text modality, leveraging its generalization ability to provide semantic compensation for visual features, enabling the model to learn cross-domain consistent representations and reduce model hallucination.
2. Through the CPTG and RCTVF modules, it generates multi-level complementary perceptive text embeddings to maximize the compensation gain for visual features.
3. While ensuring performance, CroPe demonstrates higher efficiency, achieving a good balance between efficiency and performance.

### **Weaknesses**:
1. The authors need to highlight the core contribution of CroPe. Is there a central innovation? The paper reads more like an engineering technical report. Although the process and details of the method are described clearly, it does not clarify why each module is designed as such or what the advantages of each module are.
2. The trainable parameters of the model in Figure 2 are ambiguous. Do components like the Transformer decoder, FPN, and RCTVF include trainable parameters?
3. Why does the Domain-Specific Perception module use a Transformer decoder instead of an encoder for cross-modal fusion?
4. In extremely harsh scenarios (e.g., dense fog), where visual features may severely degrade, will text semantics overly dominate, causing segmentation results to deviate from the actual scene?
5. Domain-invariant regularization relies on manually designed text prompts (e.g., "a typical driving scenario with a [class]"), whose generalizability is limited by scene diversity.

---

> ### Author Rebuttal · Authors · 2025-07-30
>
> Thanks for reviewing our work and providing these valuable comments.
>
> **W1:** Below is our clarification for all questions.
>
> 1) **Core contribution and innovation:** Our CroPe pioneers the integration of robust text modality semantics into unsupervised domain adaptive semantic segmentation (UDASS) under adverse conditions, addressing the shortcomings of existing visual-only methods in visual feature degradation (e.g., night, fog). As described in introduction section, in UDASS, visual features often suffer from severe degradation, including loss of texture, low visibility, and color distortion. In such cases, relying solely on visual prompt engineering (e.g., mask reconstruction of MIC and image style augmentation of PASS) becomes insufficient. We propose that textual semantics, which remain unaffected by environmental changes, can serve as stable high-level guidance to compensate for missing or ambiguous visual details under adverse conditions. Thus, we propose a cross-modal semantic compensation approach to enhance the generalization of visual features across diverse adverse scenarios. We validated the advantages of our modules through comprehensive experiments. For example, CroPe achieves a substantial 6.5 mIoU improvement on the ACDC-Night dataset (Table 1). Moreover, it reduces training time by 52–64%, memory usage by 28–64%, and boosts inference speed by ~45%, while enhancing scalability (Table 5).
> In summary, our method surpasses traditional visual frameworks, effectively addressing visual degradation in UDASS (e.g., preventing sky missegmentation as roads) from both theoretical and practical perspectives.
>
> 2) **Why each module is designed as such:** Our CroPe consists of two core modules, RCTVF and CPTG, for unsupervised domain adaptive semantic segmentation (UDASS) in adverse conditions. As described in Introduction section (line 36-53), the RCTVF module is designed based on the ability of text information to capture stable semantics for the same category across different domains (derived from pre-trained knowledge of CLIP), which remain invariant to visual conditions. Thus, fully integrating text semantics into multi-scale visual features effectively compensates for semantic degradation in visual features. The RCTVF module introduces a unified attention mechanism, embedding text semantics of CLIP into multi-scale visual features and refining fused features layer-by-layer via a reverse decoding chain, achieving robust cross-modal semantic compensation. The CPTG module is designed to address the risk of text prompts overfitting to the source domain and misaligning with the visual modality (line 97-103). Therefore, we use orthogonal loss to decouple domain-specific and domain-invariant semantics, with gated complementary fusion dynamically balancing both domain perception and generalization.
>
> 3) **What the advantages of each module are:** For the CPTG module, Table 2 shows that its introduction improves 1.2 mIoU, confirming that enhancing text embeddings with domain-specific perception and generalization constraints boosts model adaptability. Table 3 provides an ablation study on internal components of CPTG, further validating the effectiveness of its domain-specific perception module and generalization constraints. Table 9 demonstrates the impact of different fixed prompts on generalization, indicating the ability to adapt to various general prompts. For the RCTVF module, Table 4 verifies the superiority of the reverse decoding chain in fusion performance. Table 8 directly compares model performance with and without RCTVF, showing an average mIoU improvement of 1.6, proving the effectiveness of cross-modal fusion. Table 10 further shows that, even with only visual modality input, RCTVF outperforms existing methods (e.g., MIC), highlighting its unique value in enhancing visual feature robustness.
>
> **W2:** In Figure 2, the trainable parameters are located in the Transformer decoder, FPN, and RCTVF components. Only the CLIP text encoder, which is explicitly marked with a “frozen” icon, is not trainable.
>
> **W3:** We use a Transformer decoder in the domain-specific perception module for the following reasons:
>
> 1) Compared to cross-attention mechanism in encder, the decoder more effectively preserve the complete semantics of text prompts while dynamically integrating visual features via textual embeddings. In UDASS adverse scenarios (e.g., night, fog), the decoder leverages prompt knowledge (e.g., class semantics) to robustly compensate for visual degradation, while the encoder's cross-attention risks disrupting prompt semantic consistency.
> 2) To ensure a fair comparison and align with established practices in prior work (e.g., [1-5]), we adopt a Transformer decoder in the Domain-Specific Perception module for effective cross-modal fusion.
>
> For further validation, we compare the decoder and encoder on the CS to DZ and CS to ACDC datasets. The decoder architecture outperforms the encoder by an average of 2.1 mIoU.
> | Methods | CS to DZ | CS to ACDC-All | CS to ACDC-Night | CS to ACDC-Fog | CS to ACDC-Rain | CS to ACDC-Snow |
> |:--:|:--:|:--:|:--:|:--:|:--:|:--:|
> | Decoder |  48.7|  71.5| 54.2 | 78.2 | 72.1 | 73.0 |
> | Encoder |  47.0|  69.5|  50.5 | 76.5 | 70.9 | 70.5 |
>
> [1]Image segmentation using text and image prompts, CVPR 2022.
> [2]Segment anything, ICCV 2023.
> [3]Per-pixel classification is not all you need for semantic segmentation, NeurIPS 2021.
> [4]Textual query-driven mask transformer for domain generalized segmentation, ECCV 2024.
> [5]Masked-attention mask transformer for universal image segmentation, CVPR 2022.
>
> **W4:** In extremely harsh scenarios (e.g., dense fog), visual features may be severely degraded, and text embeddings may contribute more during fusion. However, CroPe ensures segmentation results do not significantly deviate from the actual scene through the Domain-Specific Perception module before text-visual fusion. This module aligns text embeddings with scene-specific information (e.g., texture, lighting) via an attention mechanism, enabling text embeddings to perceive the current visual context and reduce modality disparity. Table 3 shows the Domain-Specific Perception module improves mIoU by 0.7 on ACDC-All, validating its efficacy in extreme scenarios. We will supplement related explanations in the revised manuscript.
>
> **W5:** First, handcrafted text prompts (e.g.,"a typical driving scenario with a [class]") are designed to constrain learnable prompts, which are prone to overfitting the source domain (labeled) during training, enhancing generalization in cross-modal semantic compensation and preventing insufficient visual semantic enhancement. The CPTG module achieves this through domain-invariant regularization $L_{align}$.
>
> Second, regarding generalizability: In Table 9 of our paper, we compare multiple handcrafted prompts, including generic "a photo of a [class]" and the more scene-specific "a typical driving scenario with a [class]". The two prompts differ by only a marginal 0.8 mIoU compared to our overall improvement of 6.5 mIoU in the ACDC-Night dataset, indicating that the generic prompt is also sufficient for achieving strong performance, **without affecting generalization**. We use the scene-specific prompt "a typical driving scenario with a [class]" because current UDASS benchmarks, especially those under adverse conditions, are almost exclusively focused on driving scenes (e.g., TPAMI 2023: Domain adaptive and generalizable network architectures and training strategies for semantic image segmentation). Hence, prompts like "a typical driving scenario..." are designed specifically to enhance performance under these conditions. For broader applicability, prompts such as "a photo of a [class]" offer more general semantic coverage and can be used for non-driving domains.

---

> > ### Author Response · Authors · 2025-08-04
> >
> > Dear Reviewer U3Dv,
> >
> > We would like to express our gratitude to you for your time and effort in reviewing our manuscript and especially providing constructive
> > comments and valuable suggestions for our manuscript. We have provided detailed responses to the concerns raised.
> >
> > We would greatly appreciate your feedback on whether we have fully addressed your concerns. Your insights are valuable to us and will help improve the quality of our paper.
> >
> > Thank you once again for your time and consideration.
> >
> > Best regards,
> >
> > The authors of CroPe

---

> > > ### Comment · Reviewer_U3Dv · 2025-08-07
> > >
> > > Thanks for the author's response. Given that the method is more engineering-oriented, I tend to maintain the original score.

---

> > > > ### Author Response · Authors · 2025-08-07
> > > >
> > > > We sincerely appreciate your valuable comments. Please allow us to further clarify the theoretical property of our work.
> > > >
> > > > Our paper provides a novel multi-modality domain adaptation framework for cross-domain semantic segmentation, especially for the very challenging adverse conditions. In our framework, we propose the complementary-perceptive text generation module which is comprised of orthogonality constraint decoupling strategy, domain-specific perception and domain-invariant regularization. Moreover, we design the reverse chain text-visual fusion which can embed the semantics into the visual representation. Our multi-modality domain adaptation framework can provide inspiration for other cross-domain tasks in computer vision.
> > > >
> > > > We will revise the writing style of the paper, in order to present more theoretical validations and analytical aspects, so as to address your concerns.
> > > >
> > > > Thank you once again for your time and effort.

---

### Official Review · Reviewer_xJhp · 2025-07-02

**Clarity:** 3
**Significance:** 3
**Originality:** 3
**Rating:** 4
**Confidence:** 3

**Summary:**

This paper proposes a cross-modal semantic compensation adaptation method for robust scene understanding in adverse conditions (fog, snow, and night). Existing approaches only rely solely on visual information, CroPe introduces textual information to provide consistent representations. CroPe includes CPTG to provide text embeddings, RCTVF to conduct cross-modal compensation. Experiments show the effectiveness.

**Questions:**

1. It is not clear why the textual information can provide 'natural generalization ability' (line37-38) for adverse scenarios as texts only provide class information without fog, snow, or night information.
2. How to provide pseudo label information? Does it keep the same as baselines?
3. The paper lacks the ablation about L_{orth} and L_{align} to prove whether the decouple and domain-invariant regularization is essential.

**Ethical Concerns:**

["NO or VERY MINOR ethics concerns only"]

**Final Justification:**

I have carefully considered the paper and the authors' rebuttal. The authors have addressed my concerns. After thorough evaluation, I maintain my recommendation of Borderline Accept (4).

**Limitations:**

See questions.

**Paper Formatting Concerns:**

Figure 4, 7 seems unclear

**Quality:**

3

**Strengths And Weaknesses:**

Strength:
1. CroPe introduces textual information to enhance performance in adverse conditions and experiments show the effectiveness.
2. CroPe provides sufficient experiments to prove the effectiveness.

Weakness:
1. Introducing textual information is common for several segmentation tasks, the authors borrow this to adverse conditions, but the motivation and reason seems not clear and strong enough. See questions.

---

> ### Author Rebuttal · Authors · 2025-07-30
>
> Many thanks for reviewing our work and providing these valuable feedback.
>
> **W1:** First, although introducing textual information is common in segmentation tasks (e.g., referring segmentation, open-vocabulary segmentation), our motivation for introducing text in the context of unsupervised domain adaptation under adverse conditions (UDASS) is totally different from them. As described in Introduction section, our CroPe introduces text modality to offer semantic compensation for visual features in adverse scenes (e.g., fog, snow, nighttime). In UDASS, visual features often suffer from severe degradation, including loss of texture, low visibility, and color distortion. In such cases, relying solely on visual cues becomes insufficient. We propose that textual semantics, which remain unaffected by environmental changes, can serve as stable high-level guidance to compensate for missing or ambiguous visual details under adverse conditions. Second, our innovative design of complementary fusion prompts and semantic compensation underscores its uniqueness, with our motivation robustly validated by experimental results. On ACDC-Night, where visual degradation is most severe, our method achieves +6.5 mIoU over existing works.
> We will further emphasize these motivations in the revised version to enhance clarity. Thanks again for your insightful feedback.
>
> **Q1:** The goal of the task is to precisely provide the semantic labels for the target pixels in different conditions, e.g. fog, snow, nighttime. Visual features are sensitive to changes in illumination, texture, and occlusion patterns, while text providing class information ensures semantic stability across conditions and domains. In our methods, we design to provide a textual prompt for each class in an image and get new SOTA  performance. Therefore, we mention that the textual prompt can provide 'natural generalization ability', as the textual information captures the semantics for the same class in different domains which is invariant to visual conditions. The stable textual features offer semantic compensation for visual features in cross-domain settings, thereby improving the generalization ability of cross-domain sementic segmentation.
>
> **Q2:** The pseudo-labels of our method are generated following a standard self-training strategy, which is consistent with the baselines.Specifically, we use a teacher model to produce prediction probabilities, which are then converted into one-hot labels as pseudo-labels. This process is identical to the baselines, ensuring the fairness. We will clarify this in the Experiment section of revised manuscript.
>
> **Q3:** $L_{orth}$ is pivotal to the decoupling strategy, enabling MLPs to effectively decompose text embeddings $T$ into domain-invariant and domain-specific components; without it, decoupling fails, making separate ablation unnecessary. $L_{align}$ enhances the generalization of learnable prompts during training through generalizable handcrafted prompts (e.g., "a photo of a [class]"); omitting $L_{align}$ removes this critical supervision signal. To further address your concerns , we conduct supplementary experiments confirming the essential roles of $L_{orth}$ and $L_{align}$ in the CPTG module.
>
> | Methods | CS to DZ | CS to ACDC-All | CS to ACDC-Night | CS to ACDC-Fog | CS to ACDC-Rain | CS to ACDC-Snow |
> |:--:|:--:|:--:|:--:|:--:|:--:|:--:|
> | CroPe |  48.7|  71.5| 54.2 | 78.2 | 72.1 | 73.0 |
> | w/o $L_{orth}$ | 47.7 | 68.7 | 51.3 | 73.6 | 68.6 | 67.9 |
> | w/o $L_{align}$ | 48.5 | 70.1 | 52.8 | 74.9 | 69.9 | 71.6 |
>
> The results demonstrate that removing $L_{orth}$ leads to a 2.8 mIoU drop on the ACDC-All dataset, as MLPs fail to decouple, causing redundant and ineffective parameter learning. Likewise, removing $L_{align}$ eliminates the generalization constraint provided by handcrafted prompts (e.g., "a photo of a [class]"), resulting in a 1.4 mIoU drop. These findings validate the critical role of our design. We will integrate these analysis into the revised manuscript.
>
> **Paper Formatting Concerns:** Thank you for pointing out the formatting issue. Upon careful review, we found that the image resolution was reduced during the conversion process, resulting in unclear visual quality. We have re-exported high-resolution versions of the affected figures and will update them in the revised manuscript to ensure clarity.

---

> > ### Comment · Reviewer_xJhp · 2025-08-05
> >
> > Thank you for the detailed and thoughtful response. I truly appreciate the effort you've put into addressing my concerns. I have no further questions at this time.

---

> > > ### Author Response · Authors · 2025-08-05
> > >
> > > Dear Reviewer xJhp,
> > >
> > > Thank you for your valuable comments which have helped us to improve our work.
> > >
> > > Best regards,
> > >
> > > The authors of CroPe

---

### Note · Authors · 2025-08-12

We would like to thank all the reviewers for their insightful and constructive comments regarding our work. We have responded actively and successfully addressed the majority of concerns put forward by the reviewers.

Reviewer xJhp: CroPe introduces textual information to enhance performance in adverse conditions and experiments show the effectiveness. CroPe provides sufficient experiments to prove the effectiveness.
We have addressed the concerns raised by Reviewer xJhp and would like to thank Reviewer xJhp again.

Reviewer jRMA: The paper introduces CroPe, a novel Cross-Modal Semantic Compensation Adaptation framework designed for unsupervised domain adaptation (UDA) in adverse scene understanding tasks (e.g., fog, snow, rain, night). Unlike previous visual-only approaches, CroPe incorporates textual semantics via a vision-language framework to compensate for degraded visual features under challenging conditions.
We have addressed the majority of concerns raised by Reviewer jRMA. For the rest questions raised by Reviewer jRMA about our exclusive use of low-resolution images in the experiments, as our intention was to demonstrate that the proposed method achieves superior performance even under low-resolution conditions. Accordingly, we will add results from high-resolution into the revised manuscript. Also, we will supplement the performance in normal scenarios in the revised manuscript as suggest by Reviewer jRMA.
We would like to thank Reviewer jRMA again.

Reviewer p1zo: The methodology figure is well-made and clearly demonstrated the key designs of the proposed framework. Experiments contain comprehensive comparisons with various baseline models and under different adverse weather settings.
We have addressed some concerns of p1zo and thanks for his increasing the rating to 4. Accordingly, we will cite proper references to support our claims and add the relevant classes in the appendix to provide supplementary information as recommended.
We would like to thank Reviewer p1zo again.

Reviewer U3Dv: CroPe introduces the text modality, leveraging its generalization ability to provide semantic compensation for visual features, enabling the model to learn cross-domain consistent representations and reduce model hallucination.
We wish to highlight that our paper proposes a novel multi-modality domain adaptation framework for cross-domain semantic segmentation, which is a fundamental problem in computer vision.
We would like to thank Reviewer U3Dv again.

---

### Decision · Program_Chairs · 2025-09-17

**Decision:**

Accept (poster)

**Comment:**

This paper proposes a cross-modal semantic compensation adaptation method for robust scene understanding in adverse conditions (fog, snow, and night). Existing approaches only rely solely on visual information, CroPe introduces textual information to provide consistent representations. CroPe includes CPTG to provide text embeddings, RCTVF to conduct cross-modal compensation. Experiments show the effectiveness.

Strength:
- CroPe introduces textual information to enhance performance in adverse conditions, and experiments show the effectiveness.
- The experimental results are decent.
- The paper is well-written and easy to follow.
- The authors provide additional quantitative results to deal with questions raised by reviewers, including ablations, encoder vs. decoder in the domain-specific perception module, and experiments with different prompts.

Weakness:
- The core idea and innovation should be discussed more in the Introduction section, as many reviewers raise questions about the novelty.
- The mentioned experiments during the author rebuttal phase should be added in the revised version, i.e., experiments on normal scenarios.